# Intrinsic dynamics of randomly clustered networks generate place fields and preplay of novel environments

**Jordan Breffle[1], Hannah Germaine[1], Justin D Shin[1,2,3], Shantanu P Jadhav[1,2,3], Paul Miller[1,2,4]***

[1]Neuroscience Program, Brandeis University, Waltham, United States; [2]Volen National Center for Complex Systems, Brandeis University, Waltham, United States; [3]Department of Psychology , Brandeis University, Waltham, United States; [4]Department of Biology, Brandeis University, Waltham, United States

**\*For correspondence:**
pmiller@brandeis.edu

**Competing interest:** The authors declare that no competing interests exist.

**Abstract** During both sleep and awake immobility, hippocampal place cells reactivate time-compressed versions of sequences representing recently experienced trajectories in a phenomenon known as replay. Intriguingly, spontaneous sequences can also correspond to forthcoming trajectories in novel environments experienced later, in a phenomenon known as preplay. Here, we present a model showing that sequences of spikes correlated with the place fields underlying spatial trajectories in both previously experienced and future novel environments can arise spontaneously in neural circuits with random, clustered connectivity rather than pre-configured spatial maps. Moreover, the realistic place fields themselves arise in the circuit from minimal, landmark-based inputs. We find that preplay quality depends on the network's balance of cluster isolation and overlap, with optimal preplay occurring in small-world regimes of high clustering yet short path lengths. We validate the results of our model by applying the same place field and preplay analyses to previously published rat hippocampal place cell data. Our results show that clustered recurrent connectivity can generate spontaneous preplay and immediate replay of novel environments. These findings support a framework whereby novel sensory experiences become associated with preexisting "pluripotent" internal neural activity patterns.

## eLife assessment

This study presents an **important** finding on the spontaneous emergence of structured activity in artificial neural networks endowed with specific connectivity profiles. The evidence supporting the claims of the authors is **convincing**, providing direct comparison between the properties of the model and neural data although investigating more naturalistic inputs to the network would have strengthened the main claims. The work will be of interest to systems and computational neuroscientists studying the hippocampus and memory processes.

## Introduction

The hippocampus plays a critical role in spatial and episodic memory in mammals (*Morris et al., 1982*; *Squire et al., 2004*). Place cells in the hippocampus exhibit spatial tuning, firing selectively in specific locations of a spatial environment (*Moser et al., 2008*; *O'Keefe and Nadel, 1978*). During sleep and quiet wakefulness, place cells show a time-compressed reactivation of spike sequences corresponding to recent experiences (*Wilson and McNaughton, 1994*; *Foster and Wilson, 2006*), known as replay.

These replay events are thought to be important for memory consolidation, often referred to as memory replay (*Carr et al., 2011*).

The CA3 region of the hippocampus is a highly recurrently connected region that is the primary site of replay generation in the hippocampus. Input from CA3 supports replay in CA1 (*Csicsvari et al., 2000*; *Yamamoto and Tonegawa, 2017*; *Nakashiba et al., 2008*; *Nakashiba et al., 2009*), and peri-ripple spiking in CA3 precedes that of CA1 (*Nitzan et al., 2022*). The recurrent connections support intrinsically generated bursts of activity that propagate through the network.

Most replay models rely on a recurrent network structure in which a map of the environment is encoded in the recurrent connections of CA3 cells, such that cells with nearby place fields are more strongly connected. Some models assume this structure is pre-existing (*Haga and Fukai, 2018*; *Pang and Fairhall, 2019*), and some show how it could develop over time through synaptic plasticity (*Theodoni et al., 2018*; *Jahnke et al., 2015*). Related to replay models based on place-field distance-dependent connectivity is the broader class of synfire-chain-like models. In these models, neurons (or clusters of neurons) are connected in a one-dimensional feed-forward manner (*Diesmann et al., 1999*; *Chenkov et al., 2017*). The classic idea of a synfire-chain has been extended to included recurrent connections, such as by *Chenkov et al., 2017*; however, such models still rely on an underlying one-dimensional sequence of activity propagation.

A problem with these models is that in novel environments place cells remap immediately in a seemingly random fashion (*Leutgeb et al., 2005*; *Muller et al., 1987*). The CA3 region, in particular, undergoes pronounced remapping (*Leutgeb et al., 2004*; *Leutgeb et al., 2005*; *Alme et al., 2014*). A random remapping of place fields in such models that rely on environment-specific recurrent connectivity between place cells would lead to recurrent connections that are random with respect to the novel environment, and thus would not support replay of the novel environment.

Rather, these models require a pre-existing structure of recurrent connections to be created for each environment. A proposed solution to account for remapping in hippocampal models is to assume the existence of multiple independent and uncorrelated spatial maps stored within the connections between cells. In this framework, the maximum number of maps is reached when the noise induced via connections needed for alternative maps becomes too great for a faithful rendering of the current map (*Samsonovich and McNaughton, 1997*; *Battaglia and Treves, 1998*; *Azizi et al., 2013*). However, experiments have found that hippocampal representations remain uncorrelated, with no signs of representation re-use, after testing as many as 11 different environments in rats (*Alme et al., 2014*).

Rather than re-using a previously stored map, another possibility is that a novel map for a novel environment is generated *de novo* through experience-dependent plasticity while in the environment. Given the timescales of synaptic and structural plasticity, one might expect that significant experience within each environment is needed to produce each new map. However, replay can occur after just 1–2 laps on novel tracks (*Foster and Wilson, 2006*; *Berners-Lee et al., 2022*), which means that the synaptic connections that allow the generation of the replayed sequences must already be present. Consistent with this expectation, it has been found that decoded sequences during sleep show significant correlations when decoded by place fields from future, novel environments. This phenomenon is known as preplay and has been observed in both rodents (*Dragoi and Tonegawa, 2011*; *Dragoi and Tonegawa, 2013*; *Grosmark and Buzsáki, 2016*; *Liu et al., 2019*) and humans (*Vaz et al., 2023*).

The existence of both preplay and immediate replay in novel environments suggests that the preexisting recurrent connections in the hippocampus that generate replay are somehow correlated with the pattern of future place fields that arise in novel environments. To reconcile these experimental results, we propose a model of intrinsic sequence generation based on randomly clustered recurrent connectivity, wherein place cells are connected within multiple overlapping clusters that are random with respect to any future, novel environment. Such clustering is a common motif across the brain, including the CA3 region of the hippocampus (*Guzman et al., 2016*) as well as cortex (*Song et al., 2005*; *Perin et al., 2011*), naturally arises from a combination of Hebbian and homeostatic plasticity in recurrent networks (*Bourjaily and Miller, 2011*; *Litwin-Kumar and Doiron, 2014*; *Lynn et al., 2022*), and spontaneously develops in networks of cultured hippocampal neurons (*Antonello et al., 2022*).

As an animal gains experience in an environment, the pattern of recurrent connections of CA3 would be shaped by Hebbian plasticity (*Debanne et al., 1998*; *Mishra et al., 2016*). Relative to CA1, which has little recurrent connectivity, CA3 has been found to have both more stable spatial

tuning and a stronger functional assembly organization, consistent with the hypothesis that spatial coding in CA3 is influenced by its recurrent connections (*Sheintuch et al., 2023*). Gaining experience in different environments would then be expected to lead to individual place cells participating in multiple formed clusters. Such overlapping clustered connectivity may be a general feature of any hippocampal and cortical region that has typical Hebbian plasticity rules. *Sadovsky and MacLean, 2014*, found such structure in the spontaneous activity of excitatory neurons in primary visual cortex, where cells formed overlapping but distinct functional clusters. Further, such preexisting clusters may help explain the correlations that have been found in otherwise seemingly random remapping (*Kinsky et al., 2018*; *Whittington et al., 2020*) and support the rapid hippocampal representations of novel environments that are initially generic and become refined with experience (*Liu et al., 2021*). Such clustered connectivity likely underlies the functional assemblies that have been observed in hippocampus, wherein groups of recorded cells have correlated activity that can be identified through independent component analysis (*Peyrache et al., 2010*; *Farooq et al., 2019*).

Since our model relies on its random recurrent connections for propagation of activity through the network during spontaneous activity, we also sought to assess the extent to which the internal activity within the network can generate place cells with firing rate peaks at a location where they do not receive a peak in their external input. While the total input to the network is constant as a function of position, each cell only receives a peak in its spatially linearly varying feedforward input at one end of the track. Our reasoning is that landmarks in the environment, such as boundaries or corners, provide location-specific visual input to an animal, but locations between such features are primarily indicated by their distance from them, which in our model is represented by reduction in the landmark-specific input. One can therefore equate our model's inputs as corresponding to boundary cells (*Savelli et al., 2008*; *Solstad et al., 2008*; *Bush et al., 2014*), and the place fields between boundaries are generated by random internal structure within the network. Further, variations in spatial input forms do not affect the consistency and robustness of the model.

In our implementation of this model, we find that spontaneous sequences of spikes generated by a randomly clustered network can be decoded as spatial trajectories without relying on pre-configured, environment-specific maps. Because the network contains neither a preexisting map of the environment nor an experience-dependent plasticity, we refer to the spike-sequences it generates as preplay. However, the model can also be thought of as a preexisting network in which immediate replay in a novel environment can be expressed and then reinforced through experience-dependent plasticity. We find that preplay in this model occurs most strongly when the network parameters are tuned to generate networks that have a small-world structure (*Watts and Strogatz, 1998*; *Haga and Fukai, 2018*; *Humphries and Gurney, 2008*). Our results support the idea that preplay and immediate replay could be a natural consequence of the preexisting recurrent structure of the hippocampus.

## Results
### The model

We propose a model of preplay and immediate replay based on randomly clustered recurrent connections (*Figure 1*). In prior models of preplay and replay, a preexisting map of the environment is typically assumed to be contained within the recurrent connections of CA3 cells, such that cells with nearby place fields are more strongly connected (*Figure 1a*). While this type of model successfully produces replay (*Haga and Fukai, 2018*; *Pang and Fairhall, 2019*), such a map would only be expected to exist in a familiar environment, after experience-dependent synaptic plasticity has had time to shape the network (*Theodoni et al., 2018*). It remains unclear how, in the absence of such a preexisting map of the environment, the hippocampus can generate both preplay and immediate replay of a novel environment.

Our proposed alternative model is based on a randomly clustered recurrent network with random feed-forward inputs (*Figure 1b*). In our model, all excitatory neurons are randomly assigned to overlapping clusters that constrain the recurrent connectivity, and they all receive the same linear spatial and contextual input cues which are scaled by randomly drawn, cluster-dependent connection weights (see Methods). This bias causes cells that share cluster memberships to have more similar place fields during the simulated run period, but, crucially, this bias is not present during sleep simulations so that there is no environment-specific information present when the network generates preplay.

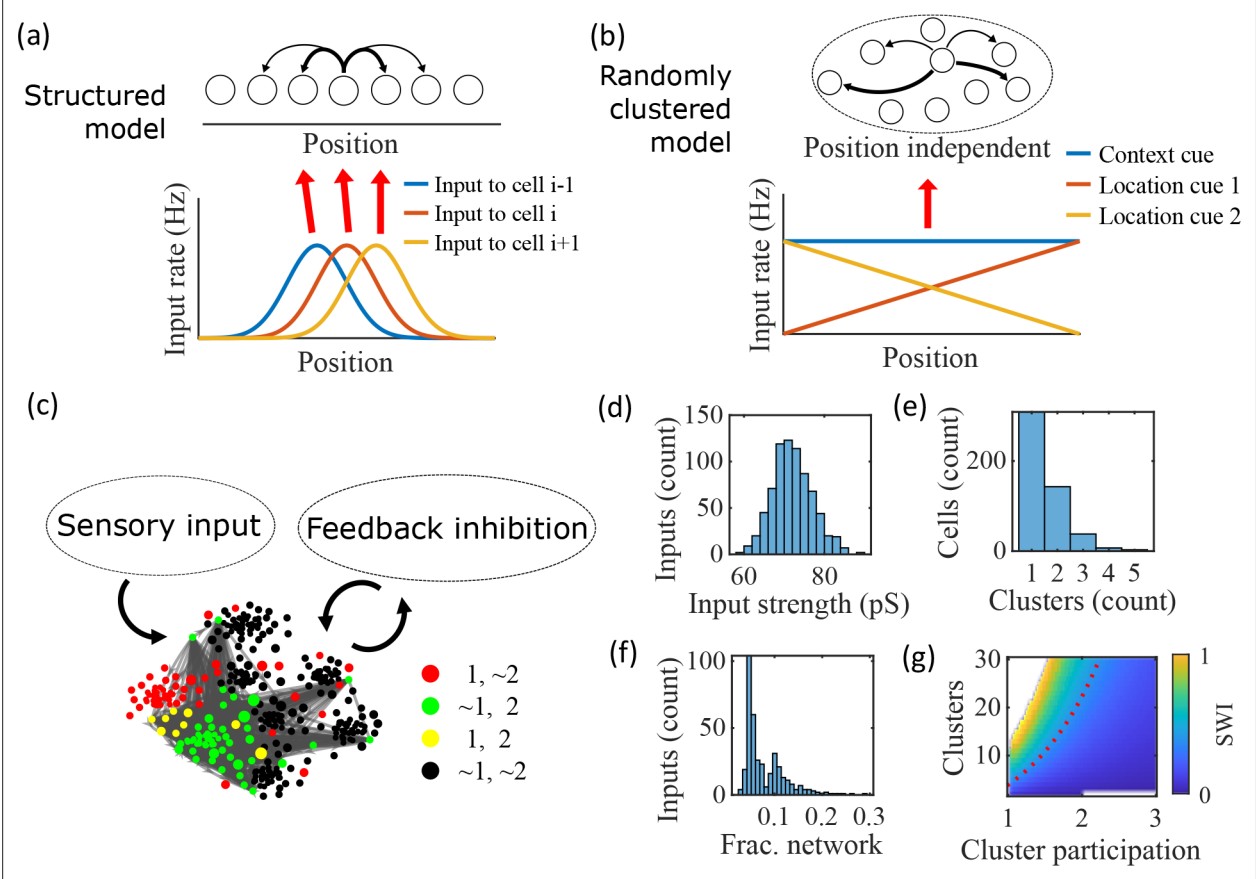

**Figure 1.** Illustration of the randomly clustered model. (**a**) Schematic diagram of prior replay models that rely on preexisting environment-specific structure, wherein each cell receives uniquely tuned Gaussian-shaped feed-forward inputs to define the place fields, and cells with nearby place fields are recurrently connected. Pairs of cells with closest place fields are connected most strongly (thicker arrows). (**b**) Schematic diagram of our model, where neurons are randomly placed into clusters and all neurons receive the same spatial and contextual information but with random, cluster-dependent input strengths. (**c**) Example representation of the network (8 clusters, mean cluster participation per cell of 1.5). Excitatory cells (each symbol) are recurrently connected with each other and with inhibitory cells ('Feedback inhibition', individual inhibitory cells not shown) and receive feed forward input ('Sensory input'). Symbol color indicates neurons' membership in clusters 1 and 2, with ~meaning not in the cluster. Symbol size scales with the number of clusters a neuron is in. Lines show connections between neurons that are in cluster 2. Symbol positions are plotted based on a t-distributed stochastic neighbor embedding (t-SNE) of the connection matrix, which reveals the randomly overlapping clusters. (**d-f**) Histograms based on the network in (**c**) of: (**d**) the distribution of input strengths; (**e**) the number of clusters that each neuron is a member of; and (**f**) the fraction of the excitatory cells to which each excitatory cell connects. (**g**) The Small-World Index (SWI) of the excitatory connections varies with the number of clusters and the mean number of clusters of which each neuron is a member ("cluster participation"). The median value of the SWI from 10 networks at each parameter point is plotted. The red dashed line shows a contour line where SWI = 0.4. Regions in white are not possible due to either cluster participation exceeding the number of clusters (lower right) or cells not being able to connect to enough other cells to reach the target global connectivity $p_c$ (upper left).

The online version of this article includes the following figure supplement(s) for figure 1:

**Figure supplement 1.** Comparison of the randomly clustered network and the canonical Watts-Strogatz small-world network.

An example network with 8 clusters and cluster participation of 1.5 (the mean number of clusters to which an excitatory neuron belongs) is depicted in *Figure 1c*. Excitatory neurons are recurrently connected to each other and to inhibitory neurons. Inhibitory cells have cluster-independent connectivity, such that all E-to-I and I-to-E connections exist with a probability of 0.25. Feed-forward inputs are independent Poisson spikes with random connection strength for each neuron (*Figure 1d*). Excitatory cells are randomly, independently assigned membership to each of the clusters in the network. All neurons are first assigned to one cluster, and then randomly assigned additional clusters to reach the target cluster participation (*Figure 1e*). Given the number of clusters and the cluster participation, the within-cluster connection probability is calculated such that the global connection probability matches the parameter $p_c = 0.08$ (*Figure 1f*). The left peak in the distribution shown in *Figure 1f* is

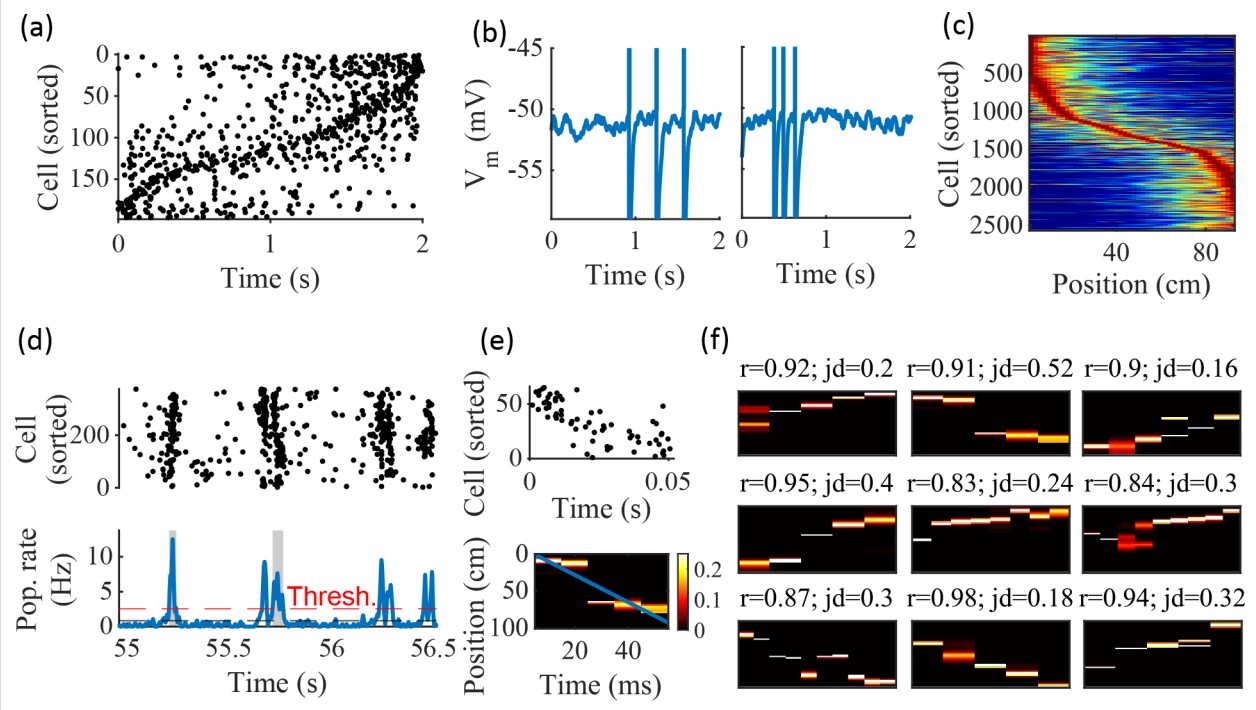

**Figure 2.** Spatially correlated reactivations in networks without environment-specific connectivity or plasticity. (**a–f**) Example activity from the fiducial parameter set (15 clusters, mean cluster participation of 1.25). (**a**) Example raster plot from one place-field trial. Cells sorted by trial peak. (**b**) Example membrane traces from two of the cells in (**a**). (**c**) Place fields from 10 different networks generated from the same parameter set, sorted by peak location and normalized by peak rate. (**d**) Example raster plot (top) and population firing rate (bottom; blue line) showing preplay in a simulation of sleep. Horizontal dashed black line is the mean population rate across the simulation. Horizontal dashed red line is the threshold for detecting a population-burst event (PBE). PBEs that exceeded the threshold for at least 50 ms and had at least five participating cells were included in the preplay decoding analysis. Grey bars highlight detected events. (**e**) Example preplay event (Top, raster plot. Bottom, Bayesian decoding of position). Event corresponds to the center event in (**d**). Raster includes only participating cells. The blue line shows the weighted correlation of decoded position across time. (**f**) Nine example decoded events from the same networks in (**c**). The width of each time bin is 10 ms. The height spans the track length. Same color scale as in (**e**). r is each event's absolute weighted correlation. jd is the maximum normalized jump in peak position probability between adjacent time bins. The same event in (**e**) is shown with its corresponding statistics in the center of the top row. Preplay statistics calculated as in *Farooq et al., 2019*.

from cells in a single cluster and the right peak is from cells in two clusters, with the long tail corresponding to cells in more than two clusters.

For a given $p_c$, excitatory connectivity is parameterized by the number of clusters in the network and the mean cluster participation. The small-world index (SWI; *Neal, 2015*; *Neal, 2017*) systematically varies across this 2-D parameterization (*Figure 1g*). A high SWI indicates a network with both clustered connectivity and short path lengths (*Watts and Strogatz, 1998*). A ring lattice network (*Figure 1—figure supplement 1a*) exhibits high clustering but long path lengths between nodes on opposite sides of the ring. In contrast, a randomly connected network (*Figure 1—figure supplement 1c*) has short path lengths but lacks local clustered structure. A network with small world structure, such as a Watts-Strogatz network (*Watts and Strogatz, 1998*) or our randomly clustered model (*Figure 1—figure supplement 1b*), combines both clustered connectivity and short path lengths. In our clustered networks, for a fixed connection probability, SWI increases with more clusters and lower cluster participation, so long as cluster participation is greater than one to ensure sparse overlap of (and hence connections between) clusters. Networks in the top left corner of *Figure 1g* are not possible, since in that region all within-cluster connections are not sufficient to match the target global connectivity probability, $p_c$. Networks in the bottom right are not possible because otherwise mean cluster participation would exceed the number of clusters. The dashed red line shows an example contour line where $SWI = 0.4$.

## Example activity

Our randomly clustered model produces both place fields and preplay with no environment-specific plasticity or preexisting map of the environment (*Figure 2*). Example place cell activity shows spatial

specificity during linear track traversal (*Figure 2a–c*). Although the spatial tuning is noisy, this is consistent with the experimental finding that the place fields that are immediately expressed in a novel environment require experience in the environment to stabilize and improve decoding accuracy (*Tang and Jadhav, 2022*; *Shin et al., 2019*; *Hwaun and Colgin, 2019*). Raster plots of network spiking activity (*Figure 2a*) and example cell membrane potential traces (*Figure 2b*) demonstrate selective firing in specific track locations. Place fields from multiple networks generated from the same parameters, but with different input and recurrent connections, show spatial tuning across the track (*Figure 2c*).

To test the ability of the model to produce preplay, we simulated sleep sessions in the same networks. Sleep sessions were simulated in a similar manner to the running sessions but with no location cue inputs active and a different, unique set of context cue inputs active to represent the sleep context. The strength of the context cue inputs to the excitatory and inhibitory cells were scaled in order to generate an appropriate level of network activity, to account for the absence of excitatory drive from the location inputs (see Methods). During simulated sleep, sparse, stochastic spiking spontaneously generates sufficient excitement within the recurrent network to produce population burst events resembling preplay (*Figure 2d–f*). Example raster and population rate plots demonstrate spontaneous transient increases in spiking that exceed 1 standard deviation above the mean population rate denoting population burst events (PBEs; *Figure 2d*). We considered PBEs that lasted at least 50 ms and contained at least five participating cells candidates for Bayesian decoding (*Shin et al., 2019*). Bayesian decoding of an example PBE using the simulated place fields reveals a spatial trajectory (*Figure 2e*). We use the same two statistics as *Farooq et al., 2019* to quantify the quality of the decoded trajectory: the absolute weighted correlation (r) and the maximum jump distance (jd; *Figure 2f*). The absolute weighted correlation of a decoded event is the absolute value of the linear Pearson's correlation of space-time weighted by the event's derived posteriors. Since sequences can correspond to either direction along the track, the sign of the correlation simply indicates direction while the absolute value indicates the quality of preplay. The maximum jump distance of a decoded event is the maximum jump in the location of peak probability of decoded position across any two adjacent 10 ms time bins of the event's derived posteriors. A high-quality event will have a high absolute weighted correlation and a low maximum jump distance.

Together, these results demonstrate that the model can reproduce key dynamics of hippocampal place cells, including spatial tuning and preplay, without relying on environment-specific recurrent connections.

## Place fields

To compare the place fields generated by the model to those from hippocampal place cells of rats, we calculated several place-field statistics for both simulated and experimentally recorded place fields (*Figure 3*). Because our model assumes no previous environment-specific plasticity, we analyzed data from place cells in rats on their first exposure to a W-track (*Shin et al., 2019*). Equivalent statistics of place-field peak rate, sparsity, and spatial information are shown for experimental data (*Figure 3a*) and simulations (*Figure 3b*). We found that the model produces qualitatively similar (but not quantitatively identical) distributions for the fiducial parameter set.

These place-field properties depend on the network parameters (*Figure 3c*). With fewer clusters and lower cluster overlap (lower cluster participation), place fields have higher peak rates, sparsity, and spatial information (*Figure 3c*, top row and bottom left). However, lower overlap reduces the uniformity of place-field locations, measured by KL-divergence (*Figure 3c* bottom middle) and the fraction of place fields in the central third of the track (*Figure 3c* bottom right).

To verify that our simulated place cells were more strongly coding for spatial location than for elapsed time, we performed simulations with additional track traversals at different speeds and compared the resulting place fields and time fields in the same cells. We find that there is significantly greater place information than time information (*Figure 3—figure supplement 1*).

## Preplay

Having found that the model produces realistic place-field representations with neither place-field like inputs nor environment-specific spatial representation in the internal network connectivity (*Figure 3*), we next examined whether the same networks could generate spontaneous preplay of novel environments. To test this, for the same set of networks characterized by place-field properties in *Figure 3*,

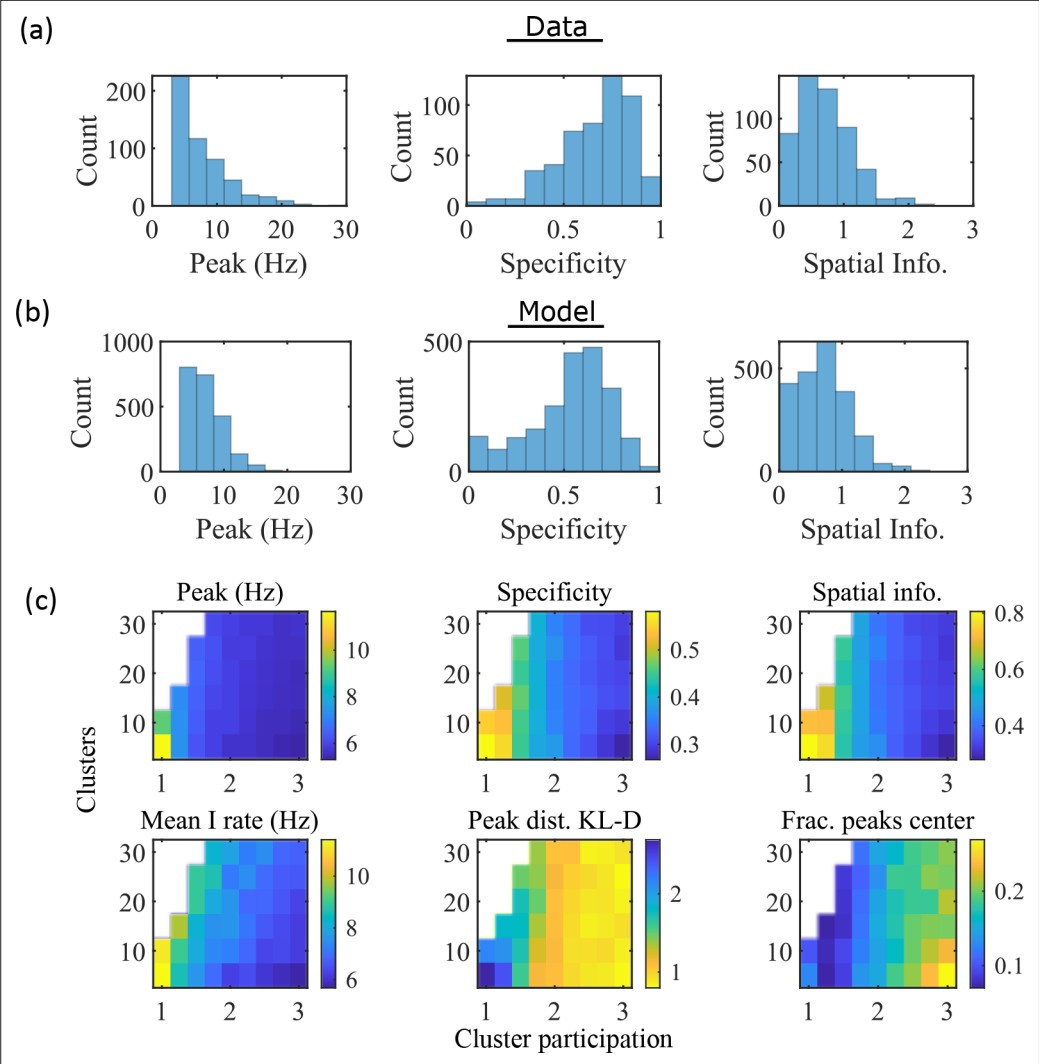

**Figure 3.** The model produces place fields with similar properties to hippocampal place fields. (**a**) Place field statistics for hippocampal place fields recorded in rats upon their first exposure to a W-track (***Shin et al., 2019***). Left, place-field peak rate (Hz). Center, place-field specificity (fraction of track). Right, place-field spatial information (bits/spike). (**b**) Same as (**a**) but for place fields from a set of 10 simulated networks at one parameter point (15 clusters and mean cluster participation of 1.25). (**c**) Network parameter dependence of place-field statistics. For each parameter point, the color indicates the mean over all place fields from all networks. Top row: mean statistics corresponding to the same measures of place fields used in panels (**a, b**). Bottom left: mean firing rate of the inhibitory cells. Bottom center: the KL-divergence of the distribution of place-field peaks relative to a uniform spatial distribution. Bottom right: fraction of place-field peaks peaked in the central third of the track.

The online version of this article includes the following figure supplement(s) for figure 3:

**Figure supplement 1.** The simulated cells have greater place information than time information.

we simulated sleep activity by removing any location-dependent input cues and analyzed the resulting spike patterns for significant sequential structure resembling preplay trajectories (***Figure 4***). We find significant preplay in both our reference experimental data set (***Shin et al., 2019***; ***Figure 4a and b***; see ***Figure 4—figure supplement 1*** for example events) and our model (***Figure 4c and d***) when analyzed by the same methods as ***Farooq et al., 2019***, wherein the significance of preplay is determined relative to time-bin shuffled events (see Methods). The distribution of absolute weighted correlations of actual events was significantly greater than the distribution of absolute weighted correlations of shuffled events for both the experimental data (***Figure 4a***, KS-test, $p = 2 \times 10^{-12}$, KS-statistic=0.078) and the simulated data (***Figure 4c***, KS-test, $p = 3 \times 10^{-16}$, KS-statistic=0.29). Additionally, we found that this

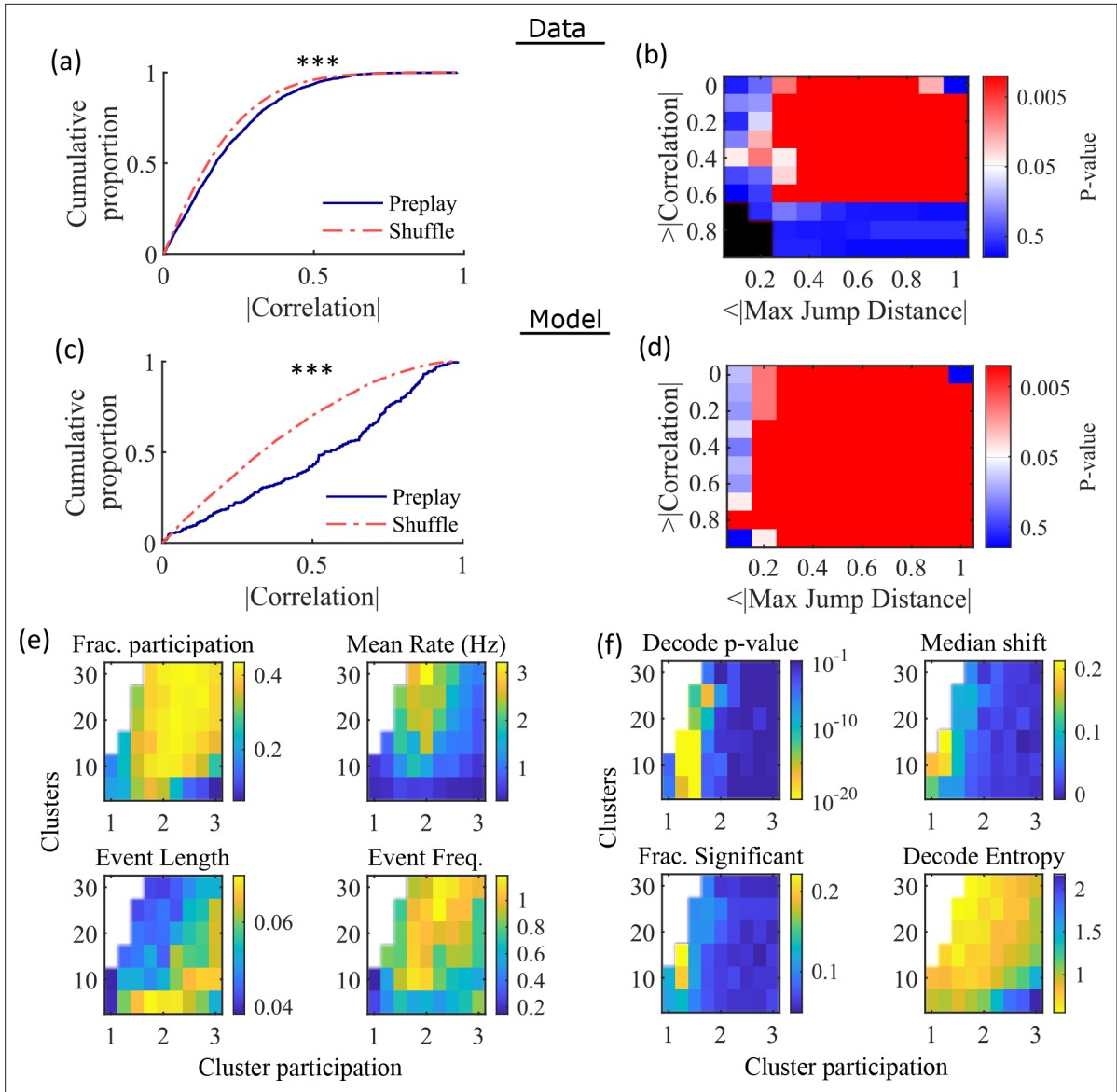

**Figure 4.** Preplay depends on modest cluster overlap. (**a, c**) The cumulative distribution function (CDF) of the absolute weighted correlations for actual events (blue line) versus shuffled events (red dashed line) of experimental data from *Shin et al., 2019* (a; KS-test, p=2 × 10⁻¹², KS-statistic=0.078) and simulated data (c; KS-test, p=3 × 10⁻¹⁶, KS-statistic=0.29) reveal results similar to those in Figure 1h of *Farooq et al., 2019*. *** p<0.001. (**b, d**) p-value grids (p-value indicated logarithmically by color) showing that the actual decoded events are higher quality sequences than shuffles across a wide range of quality thresholds for both experimental data from *Shin et al., 2019* (**b**) and simulated data (**d**). For each point on the grid, the fraction of events that exceed the absolute weighted correlation threshold (y-axis) and don't exceed the maximum jump distance (x-axis) is calculated, and the significance of this fraction is determined by comparison against a distribution of corresponding fractions from shuffled events. Black squares indicate criteria that were not met by any events (either shuffled or actual). The panel is equivalent to Figure 1e of *Farooq et al., 2019*. (**e**) Network parameter dependence of several statistics quantifying the population-burst events. Top left, fraction of excitatory cells firing per event. Top right, mean excitatory cell firing rate (Hz). Bottom left, mean event duration (s). Bottom right, mean event frequency (Hz). Each point is the mean of data combined across all population-burst events of all networks at each parameter point. Data from the same simulations as *Figure 3*. (**f**) Network parameter dependence of several statistics quantifying the Bayesian decoding. Top left, p-value of the absolute weighted correlations (from a KS-test as calculated in (**c**)). Top right, the shift in the median absolute weighted correlation of actual events relative to shuffle events. Bottom left, the fraction of events with significant absolute weighted correlations relative to the distribution of absolute weighted correlations from time bin shuffles of the event. Bottom right, the mean entropy of the position probability of all time bins in decoded trajectories.

The online version of this article includes the following figure supplement(s) for figure 4:

**Figure supplement 1.** Example preplay events from the *Shin et al., 2019* data.

**Figure supplement 2.** Significant preplay can typically be identified with as few as 50 cells.

*Figure 4 continued on next page*

Figure 4 continued

**Figure supplement 3.** Preplay statistics by trajectory for *Shin et al., 2019* data.

**Figure supplement 4.** Additional simulations support the consistency and robustness of the model to variations in spatial input forms.

result is robust to random subsampling of cells in our simulated data (*Figure 4—figure supplement 2*). Our analyses of the hippocampal data produce similar results when analyzing each trajectory independently (*Figure 4—figure supplement 3*).

For each event, we also calculated the maximum spatial jump of the peak probability of decoded position between any two adjacent time bins as a measure of the continuity of the decoded trajectory. The absolute weighted correlation (high is better) and maximum jump (low is better) were then two different measures of the quality of a decoded trajectory. We performed a bootstrap test that took both of these measures into account by setting thresholds for a minimum absolute weighted correlation and a maximum jump distance and then calculating the fraction of events meeting both criteria of quality. The significance of the fraction of events meeting both criteria was then determined by comparing it against a distribution of such fractions generated by sets of the time-bin shuffled events. We systematically varied both thresholds and found that the actual events are of significantly higher quality than chance for a wide range of thresholds in both the hippocampal (*Figure 4b*) and simulated (*Figure 4d*) data. The upper right corner of these grids cannot be significant since 100% of all possible events would be included in any shuffle or actual set. Points in the left-most column are not all significant because the strictness of the maximum jump distance means that very few events in either the actual or shuffled data sets meet the criterion, and therefore the analysis is underpowered. This pattern is similar to that seen in *Farooq et al., 2019* (as shown in their Figure 1e).

Both PBEs and preplay are significantly affected by the two network parameters (*Figure 4e and f*). The number of clusters and the extent of cluster overlap (indicated via mean cluster participation) affects PBE participation (*Figure 4e*, top left), firing rates (*Figure 4e*, top right), event durations (*Figure 4e*, bottom left), and event frequency (*Figure 4e*, bottom right). We find that significant preplay occurs only at moderate cluster overlap (*Figure 4f*, top left), where we also find the greatest increase from chance in the linearity of decoded trajectories (*Figure 4f*, top right). The fraction of events that are individually significant (determined by comparing the absolute weighted correlation of each decoded event against the set of absolute weighted correlations of its own shuffles) is similarly highest for modest cluster overlap (*Figure 4f*, bottom left). The mean entropy of position probability of each time bin of decoded trajectories is also highest for modest cluster overlap (*Figure 4f*, bottom right), meaning that high cluster overlap leads to more diffuse, less precise spatial decoding.

To test the robustness of our results to variations in input types, we simulated alternative forms of spatially modulated feedforward inputs. We found that with no parameter tuning or further modifications to the network, the model generates robust preplay with variations on the spatial inputs, including inputs of three linearly varying cues (*Figure 4—figure supplement 4a*) and two stepped cues (*Figure 4—figure supplement 4b–c*). The network is impaired in its ability to produce preplay with binary step location cues (*Figure 4—figure supplement 4d*), when there is no cluster bias (*Figure 4—figure supplement 4e*), and at greater values of cluster participation (*Figure 4—figure supplement 4f*).

## Preplay is due to successive activations of individual clusters

*Figure 4f* indicates that PBEs are best decoded as preplay when cluster participation is only slightly above one, indicating a small, but non-zero, degree of cluster overlap. We hypothesized that this can be explained as balancing two counteracting requirements: (1) Sufficient cluster overlap is necessary for a transient increase in activity in one cluster to induce activity in another cluster, so as to extend any initiated trajectory; and (2) Sufficient cluster isolation is necessary so that, early in a transient, spikes from an excited cluster preferentially add excitement to the same cluster. A network with too much cluster overlap will fail to coherently excite individual clusters—rendering decoded positions to be spread randomly throughout the track—while a network with too little cluster overlap will fail to excite secondary clusters—rendering decoded positions to remain relatively localized.

We find that the dependence of preplay on cluster overlap can indeed be explained by the manner in which clusters participate in PBEs (*Figure 5*). An example PBE (*Figure 5a*) shows transient recruitment of distinct clusters, with only one cluster prominently active at a time. We

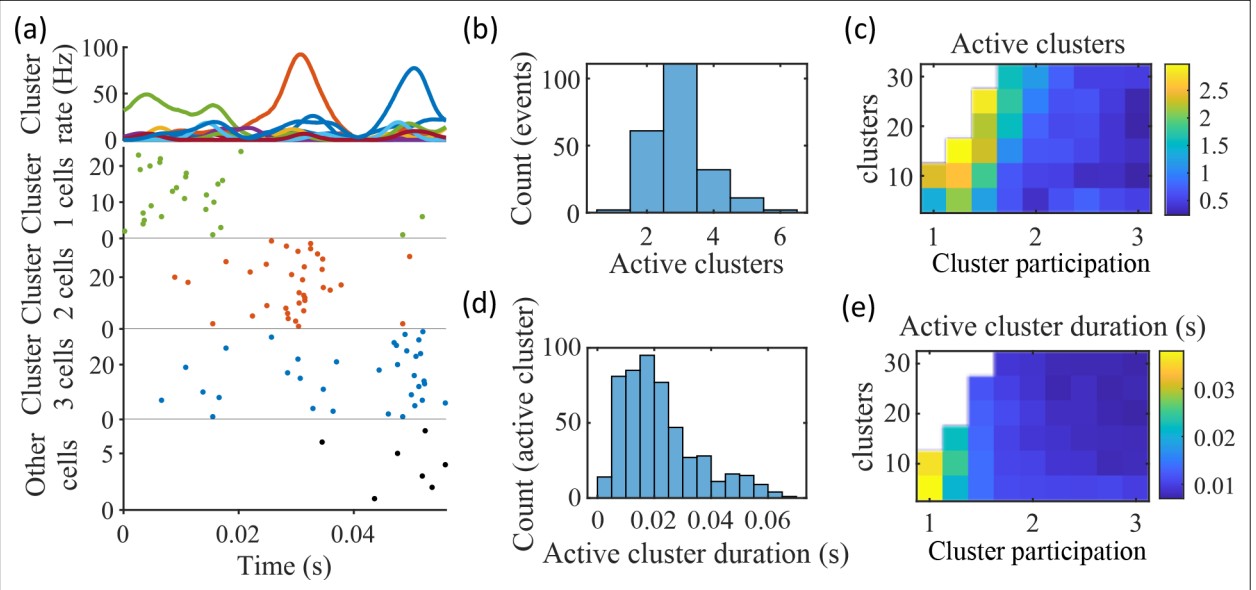

**Figure 5.** Coherent spiking within clusters supports preplay. (**a**) Example event. Top, spike rates averaged across neurons of individual clusters: Each firing rate curve is the smoothed mean firing rate across the population of cells belonging to each cluster. We defined clusters as 'active' if at any point their rates exceed twice that of any other cluster. Three clusters meet the criterion of being active (green, then red, then blue). Bottom, raster plots: Cells belonging to each of the active clusters are plotted separately in the respective colors. Cells in multiple clusters contribute to multiple population curves, and cells in multiple active clusters appear in multiple rows of the raster plot. Cells that participate but are not in any active clusters are labeled 'Other cells' and plotted in black. Only active cells are plotted. (**b**) For the fiducial parameter set (15 clusters, mean cluster participation of 1.25), the distribution over events of the number of active clusters per event. (**c**) The mean number of active clusters per event as a function of the network parameters. Same data as that used for the parameter grids in earlier figures. (**d**) For the fiducial parameter set (15 clusters, mean cluster participation of 1.25), the distribution of durations of active clusters for all active cluster periods across all events. The active duration was defined as the duration for which an active cluster remained the most-active cluster. (**e**) The mean active cluster duration as a function of the network parameters.

The online version of this article includes the following figure supplement(s) for figure 5:

**Figure supplement 1.** Relationship between cluster activation and preplay.

define a cluster as 'active' if its firing rate exceeds twice the rate of any other cluster. We calculated the number of active clusters per event (*Figure 5b*) and the duration of each active cluster period (*Figure 5d*). We find that these statistics vary systematically with the network parameters (*Figure 5c and e*), in a manner consistent with the dependence of preplay on cluster overlap (*Figure 4f*). When there is modest overlap of an intermediate number of clusters, events involve sequential activation of multiple clusters that are each active sufficiently long to correspond to at least one of the time bins used for decoding (10 ms). *Figures 4 and 5* together indicate that high-quality preplay arises via a succession of individually active clusters. Such succession requires a moderate degree of cluster overlap, but this must be combined with sufficient cluster isolation to promote independent activation of just one cell assembly for the duration of each time-bin used for decoding.

The results of *Figure 5* suggest that cluster-wise activation may be crucial to preplay. One possibility is that the random overlap of clusters in the network spontaneously produces biases in sequences of cluster activation which can be mapped onto any given environment. To test this, we looked at the pattern of cluster activations within events. We found that sequences of three active clusters were not more likely to match the track sequence than chance (*Figure 5—figure supplement 1a*). This suggests that preplay is not dependent on a particular biased pattern in the sequence of cluster activation. We then asked if the number of clusters that were active influenced preplay quality. We split the preplay events by the number of clusters that were active during each event and found that the median preplay shift relative to shuffled events with the same number of active clusters decreased with the number of active clusters (Spearman's rank correlation, p=0.0019, $\rho$ =−0.13; *Figure 5—figure supplement 1b*).

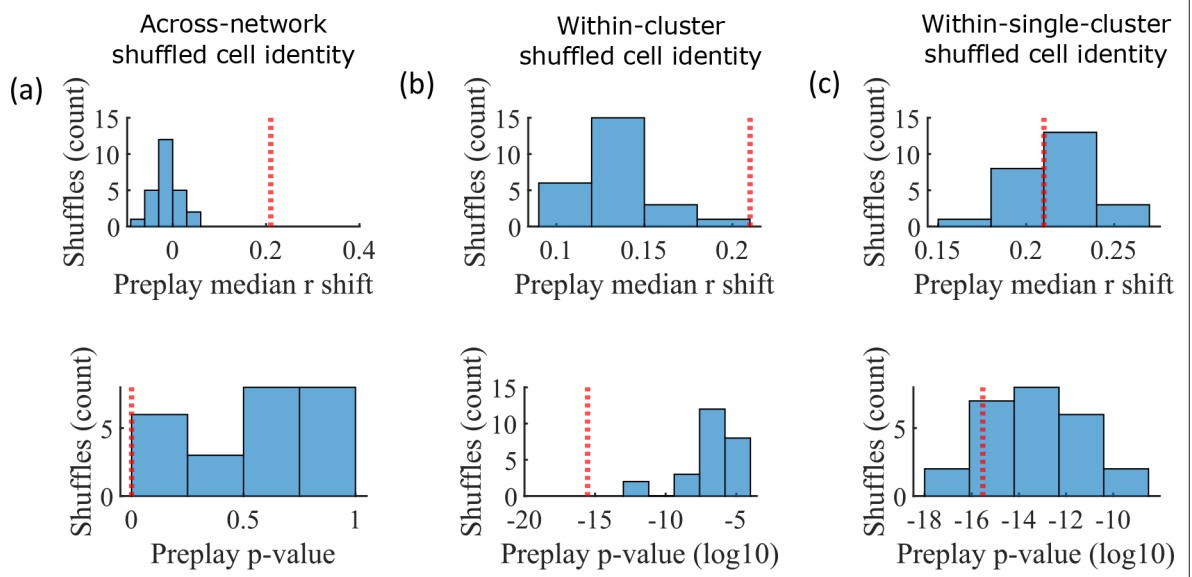

**Figure 6.** Preplay is abolished when events are decoded with shuffled cell identities but is preserved if cell identities are shuffled only within clusters. We decoded the population burst events from the fiducial parameter set simulations after randomly shuffling cell identities in three different manners (a–c, 25 replicates for each condition) and compared the resulting preplay statistics to the unshuffled result (red line). (**a**) Randomly shuffling cell identities results in median preplay correlation shifts near zero (top, 100th percentile of shuffles), with p-values distributed approximately uniformly (bottom, 0th percentile of shuffles). (**b**) Randomly shuffling cell identities within clusters reduces the magnitude of the median preplay correlation shifts (top, 100th percentile of shuffles) but preserves the statistical significance of preplay (bottom, 0th percentile of shuffles). (**c**) Randomly shuffling cell identities within clusters for only cells that belong to a single cluster results in median preplay correlation shifts that are similar to the unshuffled result (top, 36th percentile of shuffles) and are all statistically significant (bottom, 12th percentile of shuffles).

## Cluster identity is sufficient for preplay

The pattern of preplay significance across the parameter grid in *Figure 4f* shows that preplay only occurs with modest cluster overlap, and the results of *Figure 5* show that this corresponds to the parameter region that supports transient, isolated cluster-activation. This raises the question of whether cluster-identity is sufficient to explain preplay. To test this, we took the sleep simulation population burst events from the fiducial parameter set and performed decoding after shuffling cell identity in three different ways. We found that when the identity of all cells within a network are randomly permuted the resulting median preplay correlation shift is centered about zero (t-test 95% confidence interval, –0.2018–0.0012) and preplay is not significant (distribution of p-values is consistent with a uniform distribution over 0–1, chi-square goodness-of-fit test p=0.4436, chi-square statistic = 2.68; *Figure 6a*). However, performing decoding after randomly shuffling cell identity between cells that share membership in a cluster does result in statistically significant preplay for all shuffle replicates, although the magnitude of the median correlation shift is reduced for all shuffle replicates (*Figure 6b*). The shuffle in *Figure 6b* does not fully preserve cell's cluster identity because a cell that is in multiple clusters may be shuffled with a cell in either a single cluster or with a cell in multiple clusters that are not identical. Performing decoding after doing within-cluster shuffling of only cells that are in a single cluster results in preplay statistics that are not statistically different from the unshuffled statistics (t-test relative to median shift of un-shuffled decoding, p=0.1724, 95% confidence interval of –0.0028–0.0150 relative to the reference value; *Figure 6c*). Together these results demonstrate that cluster-identity is sufficient to produce preplay.

## Mean relative spike rank correlates with place field location

While cluster-identity is sufficient to produce preplay (*Figure 6b*), the shuffle of *Figure 6c* is incomplete in that cells belonging to more than one cluster are not shuffled. Together, these two shuffles leave room for the possibility that individual cell-identity may contribute to the production of preplay. It might be the case that some cells fire earlier than others, both on the track and within events. To test the contribution of individual cells to preplay, we calculated for all cells in all networks of the fiducial

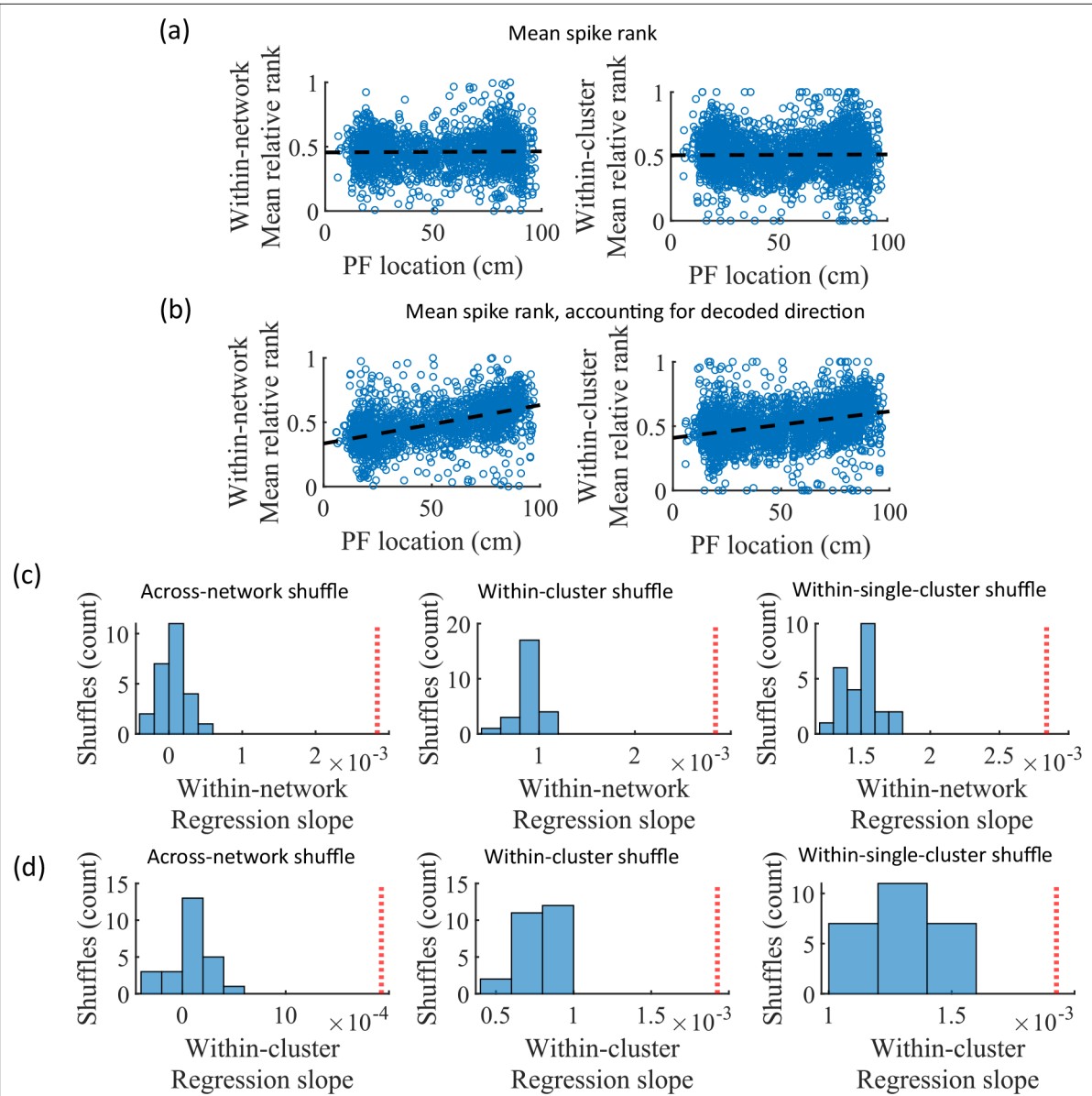

**Figure 7.** Place cells' mean event rank are correlated with their place field location when accounting for decode direction. (**a**) Mean within-event relative spike rank of all place cells as a function of the location of their mean place field density on the track for networks at the fiducial parameter set. Left, mean relative rank with respect to all cells in each network. Right, mean relative rank with respect to only cells that share cluster membership. (**b**) Same as (**a**), but after accounting for the direction of each events' decoded trajectory. If the decoded slope for a given event was negative, then the order of spiking in that event was reversed. (**c, d**) Comparison of the regression slopes from (**b**) to the distribution of slopes that results from applying the same analysis after shuffling cell identities as in *Figure 6*. (**c**) The within-network regression slope is significant relative to all three methods of shuffling cell identity. (**d**) Same as (**c**), but for the within-cluster regression slope.

parameter point their mean relative spike rank and tested if this is correlated with the location of their mean place field density on the track (*Figure 7*). We find that there is no relationship between a cell's mean relative within-event spike rank and its mean place field density on the track (*Figure 7a*). This is the case when the relative rank is calculated over the entire network (*Figure 7*, 'Within-network') and when the relative rank is calculated only with respect to cells with the same cluster membership (*Figure 7*, 'Within-cluster'). However, because preplay events can proceed in either track direction, averaging over all events would average out the sequence order of these two opposite directions. We performed the same correlation but after reversing the spike order for events with a negative slope in the decoded trajectory (*Figure 7b*). To test the significance of this correlation, we performed

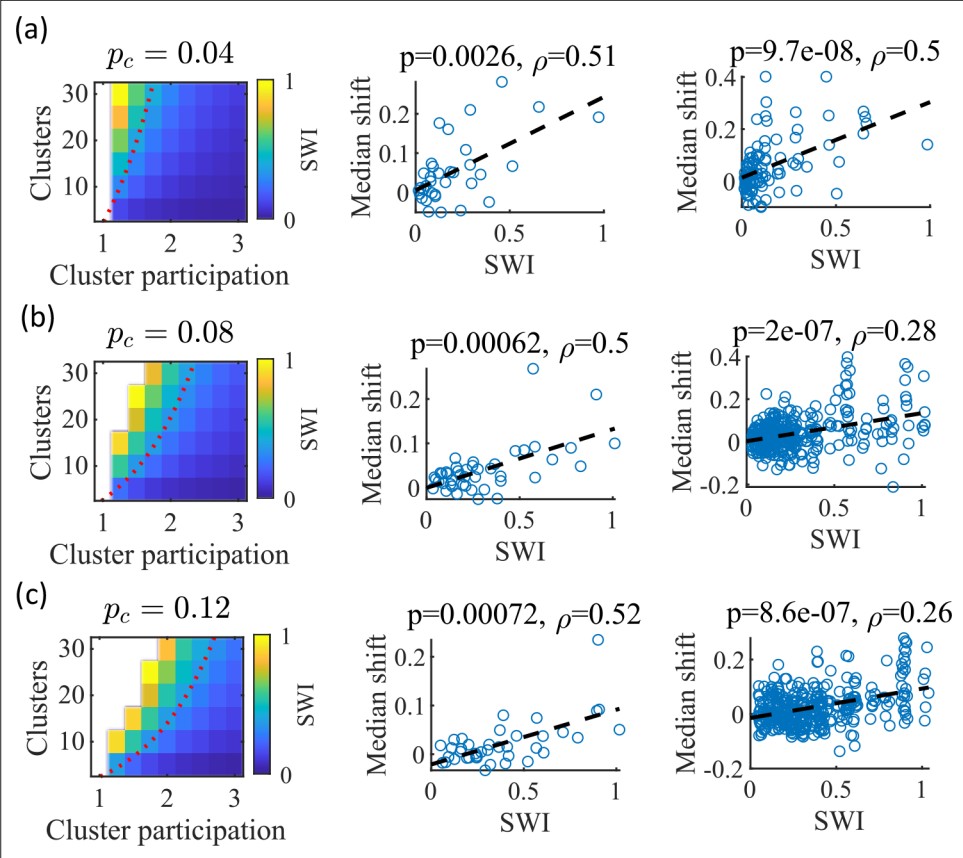

**Figure 8.** The Small-World Index of networks correlates with preplay quality. (**a–c**) Left column, the Small-World Index (SWI; plotted as color) is affected by the global E-to-E connection probability, $p_c$. Red dotted line indicates a contour line of SWI = 0.4. This boundary shifts downward as $p_c$ increases. Center column, across parameter points in the network parameter grid, SWI correlates with an increase in the median absolute weighted correlation of decoded trajectories relative to shuffles (e.g. this corresponds in *Figure 4c* to the rightward shift of the CDF of measured absolute weighted correlations relative to the shuffle events). Each point is produced by analysis of all events across 10 networks from one parameter point in the grid on the left. Right column, same as the center column but each point is data from each of the 10 individual networks per parameter set. p-value and correlation, $\rho$, are calculated from Spearman's rank-order correlation test. Dashed line is the least-squares fit. (**a**) Data from a parameter grid where the E-to-E connection probability was decreased by 50% and the E-to-E connection strength was doubled from their fiducial values used in prior figures. (**b**) Data from the same parameter grid as *Figures 3–5*. (**c**) Data from a parameter grid where the E-to-E connection probability was increased by 50% and the E-to-E connection strength scaled by two-thirds from their fiducial values.

a bootstrap significance test by comparing the slope of the linear regression to the slope that results when performing the same analysis after shuffling cell identities in the same manner as in *Figure 6*. We found that the linear regression slope is greater than expected relative to all three shuffling methods for both the within-network mean relative rank correlation (*Figure 6c*) and the within-cluster mean relative rank correlation (*Figure 6d*).

## Small-world index correlates with preplay

We noticed that that the highest quality of decoded trajectories (*Figure 4f*) seemed to arise in networks with the highest small-world index (SWI; *Figure 1g*). In order to test this, we simulated different sets of networks with both increased and decreased global E-to-E connection probability, $p_c$. Changing $p_c$, in addition to varying the number of clusters and the mean cluster participation, impacted the SWI of the networks (*Figure 8*, left column).

We hypothesized that independent of $p_c$, a higher SWI would correlate with improved preplay quality. To test this, we simulated networks across a range of parameters for three $p_c$ values: a

decrease of $p_c$ by 50% – 0.04, the fiducial value of 0.08, and an increase by 50% – 0.12 (*Figure 8a–c*, respectively). For the decreased and increased $p_c$ cases, the E-to-E connection strength was respectively doubled or reduced to 2/3 of the fiducial strength to keep total E-to-E input constant. For each parameter combination, we quantified preplay quality as the rightward shift in median absolute weighted correlation of decoded preplay events versus shuffled events (as in *Figure 4f*, top right). We then asked if there was a correlation between that quantification of preplay quality and SWI.

Across all three $p_c$ values, SWI significantly correlated with improved preplay both across parameter sets (*Figure 8*, center column) and across individual networks (*Figure 8*, right column). These results support our prediction that higher small-world characteristics correspond to higher-quality preplay dynamics regardless of average connectivity.

## Preplay significantly decodes to linear trajectories in arbitrary environments

Information about each environment enters the network via the feed-forward input connection strengths, which contain cluster-dependent biases. A new environment is simulated by re-ordering those input biases. We first wished to test that a new environment simulated in such a manner produced a distinct set of place fields. We therefore simulated place maps for leftward and rightward trajectories on linear tracks in two distinct environments (*Figure 9a*). The two maps with different directions of motion showed very high correlations when in the same environment (*Figure 9b*, blue) while the comparisons of trajectories across environments show very low correlations (*Figure 9b*, red). Cells that share membership in a cluster will have some amount of correlation in their remapping due to the cluster-dependent cue bias, which is consistent with experimental results (*Hampson et al., 1996*; *Pavlides et al., 2019*), but the combinatorial nature of cluster membership renders the overall place field map correlations low (*Figure 9b*). We also performed simulations with extra laps of running and calculated the correlations between paired sets of place fields produced by random, independent splits of trials of the same trajectory. The distribution of these correlations was similar to the distribution of within-environment correlations (comparing opposite trajectories with the same spatial input), showing no significant *de novo* place-field directionality. This is consistent with hippocampal data in which place-field directionality is initially low in novel environments and increases with experience (*Frank et al., 2004*; *Navratilova et al., 2012*; *Shin et al., 2019*).

Because we simulated preplay without any location-specific inputs, we expected that the set of spiking events that significantly decode to linear trajectories in one environment (*Figure 4*) should decode with a similar fidelity in another environment. Therefore, we decoded each PBE four times, once with the place fields of each trajectory (*Figure 9c–e*). Since the place field map correlations are high for trajectories on the same track and near zero for trajectories on different tracks, any individual event would be expected to have similar decoded trajectories when decoding based on the place fields from different trajectories in the same environment and dissimilar decoded trajectories when decoding based on place fields from different environments. A given event with a strong decoded trajectory based on the place fields of one environment would then be expected to have a weaker decoded trajectory when decoded with place fields from an alternative environment (*Figure 9c*). The distributions of absolute weighted correlations arising from decoding of PBEs according to each of the four sets of place fields was consistent across environments (*Figure 9d*, colored lines) and all were significantly rightward shifted (indicating greater absolute weighted correlation) when compared to those absolute weighted correlations arising from the corresponding shuffled events (*Figure 9d*, overlapping black lines). If we consider both absolute weighted correlation and jump-distance thresholds as in *Figure 4d*, we find that the matrices of p-values are consistent across environments (*Figure 9e*). In summary, without environment-specific or place-field dependent pre-assigned internal wiring, the model produces population-burst events, which, as an ensemble, show significant preplay with respect to any selected environment.

## Discussion

Our work shows that spontaneous population bursts of spikes that can be decoded as spatial trajectories can arise in networks with clustered random connectivity without pre-configured maps representing the environment. In our proposed model, excitatory neurons were randomly clustered with

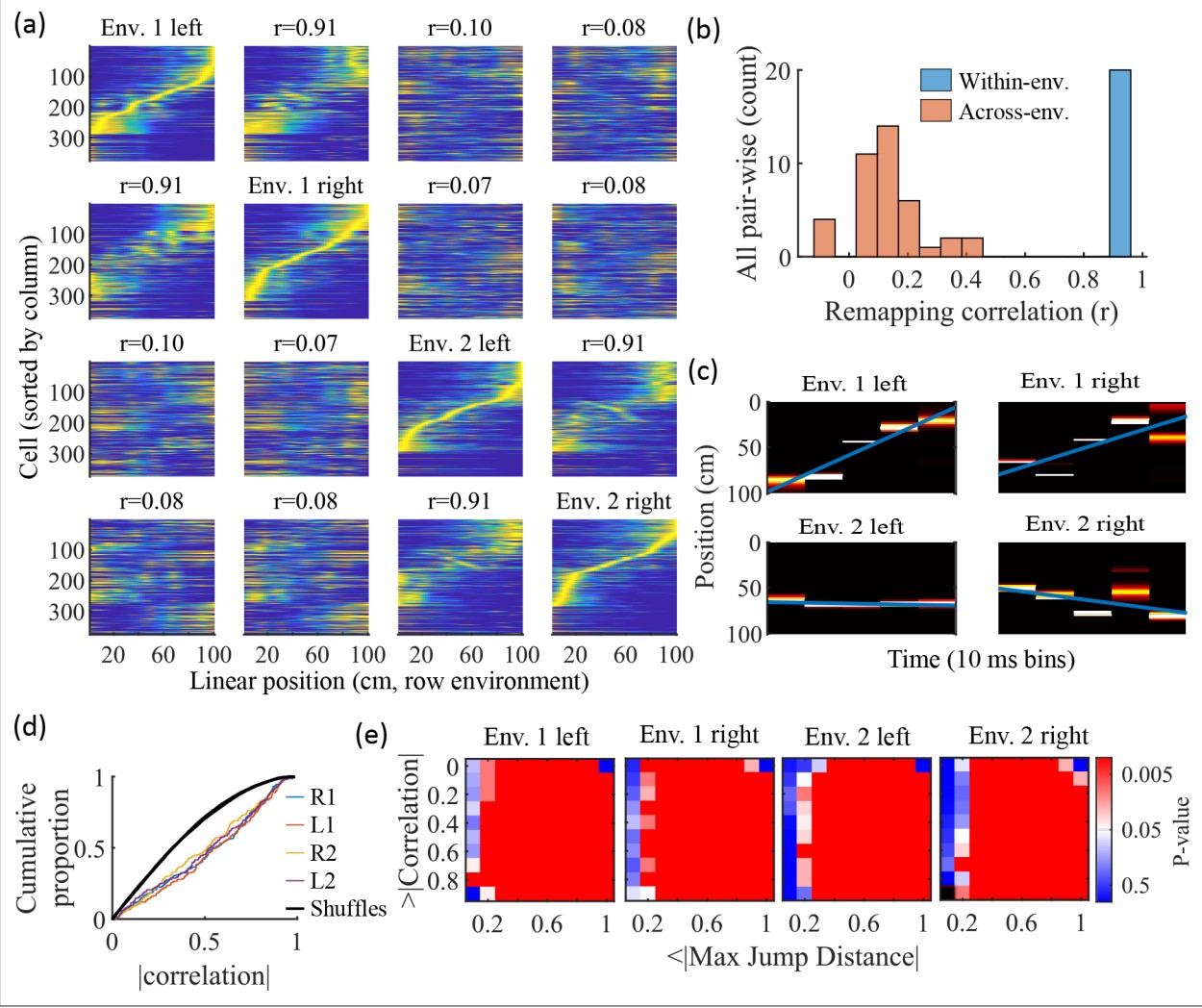

**Figure 9.** Trajectories decoded from population-burst events are significantly correlated with linear trajectories in arbitrary environments. (**a**) Place fields from a single network with simulated runs in both directions of travel on a linear track in two different environments. Each column of panels is the set of place fields for the trajectory labeled on the diagonal. Each row of panels has cells sorted by the order of place-field peaks for the trajectory labeled on the diagonal. The r values are the correlations between the corresponding remapped trajectory with its comparison on the diagonal. Note that correlations mirrored across the diagonal are equal because they correspond only to a change in the labels of the dimensions of the population rate vectors, which does not affect the vector correlation. (**b**) Distribution of the place-field map correlations across trajectories from both directions of travel on a linear track in two environments for 10 networks. Blue is the distribution of correlations for all left vs right place-field maps from the same environment. Red is the correlations from all pair-wise comparisons of trajectories from different environments. (**c**) An example event with a statistically significant trajectory when decoded with place fields from Env. 1 left (absolute correlation at the 99th percentile of time-bin shuffles) but not when decoded with place fields of the other trajectories (78th, 45th, and 63rd percentiles for Env. 1 right, Env. 2 left, and Env. 2 right, respectively). (**d**) An entire set of PBEs shows similar levels of absolute weighted correlations when decoded with different sets of place fields. In color are CDFs of absolute weighted correlations of decoded trajectories with leftward and rightward linear trajectories in each of the two environments (R1 and L1 are the rightward and leftward trajectories of environment one. R2 and L2 are the rightward and leftward trajectories of environment two). In black (all overlapping) are the corresponding absolute weighted correlations with each of the four trajectories arising from decoding of shuffled events. (**e**) The significance of linearity of decoded trajectories indicated by p-value in color (as in **Figure 4b**) from decoding the same PBEs with the four different environment place fields. Black squares indicate criteria that were not met by any events (either shuffled or actual). Env. 1 left is the same as that shown in **Figure 4d**.

varied overlap and received feed-forward inputs with random strengths that decayed monotonically from the boundaries of a track (**Figure 1**). Even though the model neural circuit lacked place-field like input and lacked environment-specific internal wiring, the network exhibited both realistic place fields (**Figures 2 and 3**) and spontaneous preplay of novel, future environments (**Figures 2 and 4**).

We validated our modeling results by applying the same analyses to a previously collected experimental data set (*Shin et al., 2019*). Indeed, we replicated the general finding of hippocampal preplay found previously in *Farooq et al., 2019*, although the p-value matrix for our experimental data (*Figure 4b*) is significant across a smaller range of threshold values than found in their prior work. This is likely due to differences in statistical power. The pre-experience sleep sessions of *Shin et al., 2019* were not longer than half an hour for each animal, while the pre-experience sleep sessions of *Farooq et al., 2019* lasted 2–4 hr. However, finding statistically significant hippocampal preplay in an experiment not designed for studying preplay shows that the general result is robust to a number of methodological choices, including shorter recording sessions, use of a W-track rather than linear track, and variations in candidate event detection criterion.

Although our model is a model of the recurrently connected CA3 region and the data set we analyze (*Shin et al., 2019*) comes from CA1 cells, the qualitative comparisons we make here are nevertheless useful. Despite some statistically significant quantitative differences, the general properties of place fields that we consider are qualitatively similar across CA1 and CA3 (*Sheintuch et al., 2023*; *Harvey et al., 2020*), and CA3 and CA1 generally reactivate in a coordinated manner (*O'Neill et al., 2008*; *Karlsson and Frank, 2009*).

The model parameters that controlled the clustering of the recurrent connections strongly influenced preplay and place-field quality. Moderate overlap of clusters balanced the competing needs for both (a) sufficiently isolated clusters to enable cluster-wise activation and (b) sufficiently overlapping clusters to enable propagation of activity across clusters (*Figure 5*). In our clustered network structure, such a balance in cluster overlap produces networks with small-world characteristics (*Watts and Strogatz, 1998*) as quantified by a small-world index (SWI; *Neal, 2015*; *Neal, 2017*). Networks with a high SWI, indicating high clustering (if two neurons are connected to the same third neuron, they are more likely than chance to be connected to each other) yet short paths (the mean number of connections needed to traverse from one neuron to any other), showed optimal preplay dynamics (*Figure 8*). The same networks could flexibly represent distinct remapped environments (*Leutgeb et al., 2004*; *Leutgeb et al., 2005*; *Alme et al., 2014*) solely through differences in scaling of feed-forward spatially linear input (*Figure 9*).

Across many species, small-world properties can be found at both the local neuronal network scale and the gross scale of the network of brain regions. At the neuronal connection scale, small-world properties have been reported in a number of networks, such as the *C. elegans* connectome (*Watts and Strogatz, 1998*; *Humphries and Gurney, 2008*), the brainstem reticular formation (*Haga and Fukai, 2018*), mouse visual cortex (*Sadovsky and MacLean, 2014*), cultured rat hippocampal neurons (*Antonello et al., 2022*), mouse prefrontal cortex (*Luongo et al., 2016*), and connectivity within the entorhinal-hippocampal region in rats (*She et al., 2016*). At the level of connected brain regions, small-world properties have been reported across the network of brain regions activated by fear memories in mice (*Vetere et al., 2017*), in the hippocampal-amygdala network in humans (*Zhang et al., 2022*), and across the entire human brain (*Liao et al., 2011*).

Our results suggest that the preexisting hippocampal dynamics supporting preplay may reflect general properties arising from randomly clustered connectivity, where the randomness is with respect to any future, novel experience. The model predicts that preplay quality will depend on the network's balance of cluster isolation and overlap, as quantified by small-world properties. Synaptic plasticity in the recurrent connections of CA3 may primarily serve to reinforce and stabilize intrinsic dynamics, which could be established through a combination of developmental programming (*Perin et al., 2011*; *Druckmann et al., 2014*; *Huszár et al., 2022*) and past experiences (*Bourjaily and Miller, 2011*), rather than creating spatial maps *de novo*. The particular neural activity associated with a given experience would then selectively reinforce the relevant intrinsic dynamics, while leaving the rest of the network dynamics unchanged.

Our model provides a general framework for understanding the origin of pre-configured hippocampal dynamics. Hebbian plasticity on independent, previously experienced place maps would produce effectively random clustered connectivity. The spontaneous dynamics of such networks would influence expression of place fields in future, novel environments. Together with intrinsic sequence

generation, this could enable preplay and immediate replay generated by the preexisting recurrent connections.

Future modeling work should explore how experience-dependent plasticity may leverage and reinforce the dynamics initially expressed through preexisting clustered recurrent connections to produce higher-quality place fields and decoded trajectories during replay (*Shin et al., 2019*; *Farooq et al., 2019*). Plasticity may strengthen connectivity along frequently reactivated spatiotemporal patterns. Clarifying interactions between intrinsic dynamics and experience-dependent plasticity will provide key insights into hippocampal neural activity. Additionally, the *in vivo* microcircuitry of CA3 is complex and includes aspects such as nonlinear dendritic computations and a variety of inhibitory cell types (*Rebola et al., 2017*). This microcircuitry is crucial for explaining certain aspects of hippocampal function, such as ripple and gamma oscillogenesis (*Ramirez-Villegas et al., 2018*), but here we have focused on a minimal model that is sufficient to produce place cell spiking activity that is consistent with experimentally measured place field and preplay statistics.

## Methods

To investigate what network properties could support preplay, we simulated recurrently connected networks of spiking neurons and analyzed their dynamics using standard hippocampal place cell analyses.

### Neuron model

We simulate networks of Leaky Integrate-and-Fire (LIF) neurons, which have leak conductance, $g_L$, excitatory synaptic conductance, $g_E$, inhibitory synaptic conductance, $g_I$, spike-rate adaptation (SRA) conductance, $g_{SRA}$, and external feed-forward input synaptic conductance, $g_{ext}$. The membrane potential, $V$, follows the dynamics

$$\tau_m \frac{dV}{dt} = -g_L \left(V - E_L\right) - g_E \left(V - E_E\right) - g_I \left(V - E_I\right) - g_{SRA} \left(V - E_{SRA}\right) - g_{ext} \left(V - E_E\right)$$

where $\tau_m$ is the membrane time constant, $E_L$ is the leak reversal potential, $E_E$ is the excitatory synapse reversal potential, $E_I$ is the inhibitory synapse reversal potential, $E_{SRA}$ is the SRA reversal potential, and $E_{ext}$ is the external input reversal potential. When the membrane potential reaches the threshold $V_{th}$, a spike is emitted and the membrane potential is reset to $V_{reset}$.

The changes in SRA conductance and all synaptic conductances follow

$$\tau_i \frac{dg_i}{dt} = -g_i$$

to produce exponential decay between spikes for any conductance $i$. A step increase in conductance occurs at the time of each spike by an amount corresponding to the connection strength for each synapse ($W_{E-E}$ for E-to-E connections, $W_{E-I}$ for E-to-I connections, and $W_{I-E}$ for I-to-E connections), or by $\delta_{SRA}$ for $g_{SRA}$. Initial feed-forward input conductances were set to values approximating their steady-state values by randomly selecting values from a Gaussian with a mean of $W_{in} r_G \tau_E$ and a standard deviation of $\sqrt{W_{in}^2 r_G \tau_E}$. Initial values of the recurrent conductances and the SRA conductance were set to zero.

| Parameter | Fiducial value | Description |
|---|---|---|
| $\tau_m$ | 40 ms | Membrane time constant |
| $C_m$ | 0.4 nF | Membrane capacitance |
| $d_t$ | 0.1 ms | Simulation time step |
| $g_L$ | 10 nS | Leak conductance |

*Continued on next page*

*Continued*

| Parameter | Fiducial value | Description |
| --- | --- | --- |
| $E_L$ | –70 mV | Leak reversal potential |
| $E_E$ | 0 mV | Excitatory synaptic reversal potential |
| $E_I$ | –70 mV | Inhibitory synaptic reversal potential |
| $E_{SRA}$ | –80 mV | SRA reversal potential |
| $V_{th}$ | –50 mV | Spike threshold |
| $V_{reset}$ | –70 mV | Reset potential |
| $\tau_E$ | 10 ms | Excitatory time constant |
| $\tau_I$ | 3 ms | Inhibitory time constant |
| $\tau_{SRA}$ | 30 ms | Spike-rate adaptation time constant |
| $\delta_{SRA}$ | 3 pS | Spike-rate adaptation strength |

## Network structure

We simulated networks of $n = 500$ neurons, of which 75% were excitatory. Excitatory neurons were randomly, independently assigned membership to each of $n_c$ clusters in the network. First, each neuron was randomly assigned membership to one of the clusters. Then, each cluster was assigned a number—$n_E (\mu_c - 1)/n_c$ rounded to the nearest integer—of additional randomly selected neurons such that each cluster had identical numbers of neurons, $n_{E,clust} = n_E (\mu_c/n_c)$, and mean cluster participation, $\mu_c$, reached its goal value.

E-to-E recurrent connections were randomly assigned on a cluster-wise basis, where only neurons that shared membership in a cluster could be connected. The within-cluster connection probability was configured such that the network exhibited a desired global E-to-E connection probability $p_c$. Given the total number of possible connections between excitatory neurons is $C_{tot} = n_E (n_E - 1)$ and the total number of possible connections between excitatory neurons within all clusters is $C_{clust} = n_{E,clust} (n_{E,clust} - 1) n_c$, we calculated the within-cluster connection probability as $p_c (C_{tot}/C_{clust})$. That is, given the absence of connections between clusters (clusters were coupled by the overlap of cells) the within-cluster connection probability was greater than $p_c$ so as to generate the desired total number of connections equal to $p_c C_{tot}$.

All E-to-I and I-to-E connections were independent of cluster membership and existed with a probability $p_{c_I}$. There were no I-to-I connections. $p_c$, $n_c$, and $\mu_c$ were varied for some simulations. Except where specified otherwise, all parameters took the fiducial value shown in the table below.

The network visualization in *Figure 1c* was plotted based on the first two dimensions of a t-distributed stochastic neighbor embedding of the connectivity between excitatory cells using the MATLAB function *tsne*. The feature vector for each excitatory cell was the binary vector indicating the presence of both input and output connections.

| Parameter | Fiducial value | Description |
| --- | --- | --- |
| $n$ | 500 | Number of neurons |
| $n_E$ | 375 | Number of excitatory neurons |
| $n_c$ or 'cluster' | 15 | Number of clusters |
| $\mu_c$ or 'cluster participation' | 1.25 | Mean cluster membership per neuron |
| $p_c$ | 0.08 | E-to-E connection probability |
| $p_{c_I}$ | 0.25 | E-to-I and I-to-E connection probability |
| $W_{E\text{-}E}$ | 220 pS | E-to-E synaptic conductance step increase |
| $W_{E\text{-}I}$ | 400 pS | E-to-I synaptic conductance step increase |
| $W_{I\text{-}E}$ | 400 pS | I-to-E synaptic conductance step increase |

## Network inputs

All excitatory neurons in the network received three different feed-forward inputs (*Figure 1b*). Two inputs were spatially modulated, with rates that peaked at either end of the track and linearly varied across the track to reach zero at the opposite end. One input was a context cue that was position independent. All excitatory cells received unique Poisson spike trains from each of the three inputs at their position-dependent rates. Inhibitory cells received only the context input.

The connection strength of each feed-forward input to each neuron was determined by an independent and a cluster-specific factor.

First, strengths were randomly drawn from a log-normal distribution $e^{\mu + \sigma \mathfrak{N}}$, where $\mathfrak{N}$ is a zero-mean, unit variance Normal distribution, $\mu = ln\left(\frac{W_{in}^2}{\sqrt{\sigma_{in} + W_{in}^2}}\right)$ and $\sigma = \sqrt{ln\left(\frac{\sigma_{in}}{W_{in}^2 + 1}\right)}$ for mean strength $W_{in}$ and standard deviation $\sigma_{in}$ for the location cues, with $\sigma_{in}$ replaced by $\sigma_{context}$ for the context cue. Each environment and the sleep session had unique context cue input weights. For model simplicity, the mean input strength $W_{in}$ for all inputs was kept the same for both E and I cells in both the awake and sleep conditions, but the strength of the resulting context input was then scaled by some factor $f_x$ for each of the four cases to accommodate for the presence, or lack thereof, of the additional current input from the location cues. These scaling factors were set at a level that generated appropriate levels of population activity. During simulation of linear track traversal, the context cue to excitatory cells was scaled down by $f_{E\text{-awake}}$ to compensate for the added excitatory drive of the location cue inputs, and the context cue input to I cells was not changed ($f_{I\text{-awake}} = 1$). During sleep simulation, the context cue input to E cells was not scaled ($f_{E\text{-awake}} = 1$) but the context cue input to I cells was scaled down by $f_{I\text{-sleep}}$.

Second, to incorporate cluster-dependent correlations in place fields, a small ($\leq 4\%$) location cue bias was added to the randomly drawn feed-forward weights based on each neuron's cluster membership. For each environment, the clusters were randomly shuffled and assigned a normalized rank bias value, such that the first cluster had a bias of –1 (corresponding to a rightward cue preference) and the last cluster had a bias of +1 (leftward cue preference). A neuron's individual bias was calculated as the mean bias of all clusters it belonged to, multiplied by the scaling factor $\sigma_{bias}$. The left cue weight for each neuron was then scaled by 1 plus its bias, and the right cue weight was scaled by 1 minus its bias. In this way, the feed-forward input tuning was biased based on the mean rank of a neuron's cluster affiliations for each environment. The addition of this bias produced correlations in cells' spatial tunings based on cluster membership, but, importantly, this bias was not present during the sleep simulations, and it did not lead to high correlations of place-field maps between environments (*Figure 9b*).

| Parameter | Value | Description |
|---|---|---|
| $r_G$ | 5000 Hz | Peak Poisson input rate |
| $W_{in}$ | 72 pS | Mean strength of the input synapses |
| $\sigma_{in}$ | 5 pS | Standard deviation of the location cue input synapses |
| $\sigma_{context}$ | 1.25 pS | Standard deviation of the context cue input synapses |
| $\sigma_{bias}$ | 0.04 | Location bias scale |
| $f_{E\text{-awake}}$ | 0.1 | E-cell context cue input scaling during awake simulation |
| $f_{E\text{-sleep}}$ | 1 | E-cell context cue input scaling during sleep simulation |
| $f_{I\text{-awake}}$ | 1 | I-cell context cue input scaling during awake simulation |
| $f_{I\text{-sleep}}$ | 0.75 | I-cell context cue input scaling during sleep simulation |

## Simulation

For a given parameter set, we generated 10 random networks. We simulated each network for one sleep session of 120 s and for five 2 s long traversals of each of the two linear trajectories on each track. For the parameter grids in *Figures 3 and 4*, we simulated 20 networks with 300 s long sleep sessions in order to get more precise empirical estimates of the simulation statistics. For analysis comparing place-field reliability, we simulated 10 traversals of each trajectory.

To compare coding for place vs time, we performed repeated simulations for the same networks at the fiducial parameter point with 1.0 x and 2.0 x of the original track traversal speed. We then combined all trials for both speed conditions to calculate both place fields and time fields for each cell from the same linear track traversal simulations. The place fields were calculated as described below (average firing rate within each of the fifty 2 cm long spatial bins across the track) and the time fields were similarly calculated but for fifty 40 ms time bins across the initial two seconds of all track traversals.

## Place field analysis

### Place-field rate maps

We followed the methods of *Shin et al., 2019* to generate place fields from the spike trains. We calculated for each excitatory cell its trial-averaged occupancy-discounted firing rate in each 2 cm spatial bin of the 1 m long linear track. Note that the occupancy-discounting term is uniform across bins, so it has no impact in our model, because we simulated uniform movement speed. We then smoothed this with a Gaussian kernel with a 4 cm standard deviation. For statistics quantifying place-field properties and for Bayesian decoding, we considered only excitatory cells with place-field peaks exceeding 3 Hz as in *Shin et al., 2019*.

### Place-field specificity

Place-field specificity was defined as 1 minus the fraction of the spatial bins in which the place field's rate exceeded 25% of its maximum rate (*Shin et al., 2019*).

### Place-field spatial information

The spatial information of each cells' place field was calculated as

$$\text{Spatial Information} = \sum_i p_i \left( \frac{r_i}{\bar{r}} \right) log_2 \left( \frac{r_i}{\bar{r}} \right)$$

where $p_i$ is the probability of being in spatial bin $i$, $r_i$ is the place field's rate in spatial bin $i$, and $\bar{r}$ is the mean rate of the place field (*Sheintuch et al., 2023*). Given the division of the track into 50 spatial bins, spatial information could vary between 0 for equal firing in all bins and $log_2(50) \cong 5.6$ for firing in only a single bin. Spatial information of 1 is equivalent, for example, to equal firing in exactly one half of the bins and no firing elsewhere.

### Distribution of peaks

We used two measures to quantify the extent to which place-field peaks were uniformly distributed across the track. In our first measure, we calculated the Kullback-Leibler divergence of the distribution of peaks from a uniform distribution, as

$$D_{KL} = -\sum_i p_i^{\text{data}} log_2 \left( \frac{p_i^{\text{uniform}}}{p_i^{\text{data}}} \right)$$

where $p_i^{data}$ is the fraction of cells with peak firing rates in the $i^{th}$ spatial bin and $p_i^{uniform}$ is 1/50, that is the fraction expected from a uniform distribution (*Sheintuch et al., 2023*). Similarly, the range for spatial information, $D_{KL}$ is bounded between zero for a perfectly uniform distribution of peaks and $log_2(50) \cong 5.6$ if all peaks were in a single bin. $D_{KL}$ of 1 is equivalent, for example, to all peaks being uniformly spread over one half of the bins in the track.

For our second measure, we calculated the fraction of place cells whose peak firing rate was in the central third of the track. Since inputs providing spatial information only peaked at the boundaries of the track, the central third was ubiquitously the most depleted of high firing rates.

### Place-field map correlations

To compare the similarity of place fields across different trajectories, we calculated the correlation between the place-field rate maps of each pair of trajectories. For each spatial bin, we calculated the Pearson correlation coefficient between the vector of the population place-field rates of the two

trajectories. We then averaged the correlation coefficients across all spatial bins to get the correlation between the two trajectories.

## PBE detection

We detected candidate preplay events in the simulated data by identifying population-burst events (PBEs). During the simulated sleep period, we calculated the mean rate of the population of excitatory cells, which defines the population rate, smoothed with a Gaussian kernel (15 ms standard deviation). We then detected PBEs as periods of time when the population rate exceeded 1 standard deviation above the mean population rate for at least 30 ms. We also required the peak population rate to exceed 0.5 Hz (corresponding to 5–6 spikes per 30 ms among excitatory cells) in order for the rate fluctuation to qualify as a PBE. We then combined PBEs into a single event if their start and end times were separated by less than 10 ms.

## Sharp-wave ripple detection

Because of the reduced number of recorded cells relative to the simulated data, we detected candidate events in the *Shin et al., 2019* data with a method that incorporated the ripple band oscillation power in the local field potential (LFP) in addition to the population spiking activity. We first calculated the smoothed firing rate for each excitatory neuron by convolving its spikes with a Gaussian kernel (100 ms standard deviation) and capping at 1 to prevent bursting dominance. We then computed the z-scored population firing rate from the capped, smoothed single-neuron rates. Additionally, we calculated the z-scored, ripple-filtered envelope of the tetrode-averaged LFP. We then summed these two z-scores and detected peaks that exceeded 6 for at least 10 ms and exceeded the neighboring regions by at least 6 (*MinPeakHeight*, *MinPeakWidth*, and *MinPeakProminence* of the MATLAB function *findpeaks*, respectively). Candidate events were defined as periods around detected peaks, spanning from when the z-score sum first dipped below 0 for at least 5 ms before the peak to after the peak when it again dipped below 0 for at least 5 ms. We additionally required that the animal be immobile during the event.

## Bayesian decoding

We performed Bayesian decoding of candidate preplay events following the methods of *Shin et al., 2019*. We performed decoding on all candidate events that had at least 5 active cells and exceeded at least 50 ms in duration. Spikes in the event were binned into 10 ms time bins. We decoded using the place fields for each trajectory independently. The description provided below is for the decoding using the place fields of one particular trajectory.

For each time bin of each event, we calculated the location on the track represented by the neural spikes based on the place fields of the active cells using a memoryless Bayesian decoder

$$P\left(x|s\right) = \frac{P\left(s|x\right) P\left(x\right)}{P\left(s\right)}$$

where $P\left(x|s\right)$ is the probability of the animal being in spatial bin $x$ given the set of spikes $s$ that occurred in the time bin, $P\left(s|x\right)$ is the probability of the spikes $s$ given the animal is in spatial bin $x$ (as given by the place fields), $P\left(x\right)$ is the prior probability of the animal being in spatial bin $x$, and $P\left(s\right)$ is the probability of the spikes $s$.

We assumed a uniform prior probability of position, $P\left(x\right)$. We assumed that the $N$ cells firing during the event acted as independent Poisson processes in order to calculate

$$P\left(s|x\right) = \prod_{i}^{N} \frac{\left(\tau r_i\left(x\right)\right)^{s_i} e^{-\tau r_i(x)}}{s_i!}$$

where $\tau$ is the time bin window duration (10 ms), $r_i\left(x\right)$ is the place-field rate of cell $i$ in spatial bin $x$ and $s_i$ is the number of spikes from cell $i$ in the time bin.

This allows us to calculate the posterior probability of position for each time bin as

$$P\left(x|s\right) = C\left(\prod_{i}^{N} r_i\left(x\right)^{s_i}\right) e^{-\tau \sum_{i}^{N} r_i(x)}$$

where $C$ is a normalization constant, which accounts for the position-independent term, $P(s)$.

## Bayesian decoding statistical analyses

We analyzed the significance of preplay using the methods of *Farooq et al., 2019* (see also *Silva et al., 2015*). We computed two measures of the sequence quality of each decoded event: the event's absolute weighted correlation and its jump distance. The absolute weighted correlation is the absolute weighted Pearson's correlation of decoded position across the event's time bins. For each decoded event, we calculate the weighted correlation between space and time with MATLAB's *fitlm* function using the decoded probability in each space-time bin (10 ms by 2 cm) as the weight for the corresponding location in the correlation. The absolute value of the weighted correlation is used in order to account for both forward and reverse preplay. The jump distance is the maximum of the distance between the positions of peak probability for any two adjacent 10 ms time bins in the event, quantified as fraction of the track length.

For each event, we generated 100 shuffled events by randomly permuting the order of the 10 ms time bins. We then calculated the weighted correlation and jump distance for each shuffled event in the same manner as for the actual events. For each simulated parameter set, we combined all events from the 10 simulated networks.

Following the methods of *Farooq et al., 2019*, we calculated the statistical significance of the population of preplay events using two different methods. First, we used the Kolmogorov-Smirnov (KS) test to compare the distributions of absolute weighted correlations obtained from the actual events and the shuffled events (*Figure 4a and c*).

Second, we used a bootstrap test to compare the fraction of high-quality events—defined as having both high absolute weighted correlations and low maximum jump distance—relative to shuffles (*Figure 4b and d*). To perform the bootstrap test, we created a grid of thresholds for minimum absolute weighted correlation and maximum jump distance, and for each combination of thresholds we calculated the fraction of actual events that exceeded the minimum absolute weighted correlation threshold and did not exceed the maximum jump distance threshold. Then, we generated 100 data sets of shuffled events by randomly permuting the order of the 10 ms time bins for each actual event and calculated the fraction of events meeting the same pairs of thresholds for each shuffled data set. The p-value of the fraction of high-quality events was then calculated as the fraction of shuffled data sets with a higher fraction of high-quality events.

To test the significance of each event's absolute weighted correlation individually, we calculated the event's p-value as the fraction of the event's own shuffles that had a higher absolute weighted correlation than the un-shuffled event (*Figure 4f*, bottom left).

The spatial entropy $H$ of a decoded event was calculated as the mean over its time bins of the entropy of the decoded position probability in each time bin, using the equation

$$H = -\sum_i p_i \, log_2\left(p_i\right)$$

for each time bin, where $p_i$ is the decoded position probability for spatial bin $i$.

## Cell identity shuffled decoding

We performed Bayesian decoding on the fiducial parameter set after shuffling cell identities in three different manners (*Figures 6 and 7*). To shuffle cells in a cluster-independent manner ('Across-network shuffle'), we randomly shuffled the identity of cells during the sleep simulations. To shuffle cells within clusters ('Within-cluster shuffle'), we randomly shuffled cell identity only between cells that shared membership in at least one cluster. To shuffle cells within only single clusters ('Within-single-cluster shuffle'), we shuffled cells in the same manner as the within-cluster shuffle but excluded any cells from the shuffle that were in multiple clusters.

To test for a correlation between spike rank during sleep PBEs and the order of place fields on the track (*Figure 7*), we calculated for each excitatory cell in each network of the fiducial parameter set its mean relative spike rank and correlated that with the location of its mean place field density on the track (*Figure 7a*). To account for event directionality, we calculated the mean relative rank after inverting the rank within events that had a negatively sloped decoded trajectory (*Figure 7b*). We calculated mean relative rank for each cell relative to all cells in the network ('Within-network mean

relative rank') and relative to only cells that shared cluster membership with the cell ('Within-cluster mean relative rank'). We then compared the slope of the linear regression between mean relative rank and place field location against the slope that results when applying the same analysis to each of the three methods of cell identify shuffles for both the within-network regression (*Figure 7c*) and the within-cluster regression (*Figure 7d*).

## Small-world index

The small-world index (SWI) was calculated following the method of *Neal, 2015* (see also *Neal, 2017*). It was defined as

$$\text{SWI} = \frac{(L - L_l)}{(L_r - L_l)} \times \frac{(C - C_r)}{(C_l - C_r)}$$

where $L$ is the mean path distance and $C$ is the clustering coefficient of the network. We calculate $L$ as the mean over all ordered pairs of excitatory cells of the shortest directed path length from the first to the second cell. We calculate $C$ as the ratio of the number of all triplets of excitatory cells that are connected in either direction over the number of all triplets that could form, following the methods of *Fagiolo, 2007* for directed graphs. $L_l$ and $C_l$ are the expected values for a one-dimensional ring lattice network with the same size and connection probability (in which connections are local such that there are no connections between cells with a greater separation on the ring than that of any pairs without a connection). And $L_r$ and $C_r$ are the expected values for a random network of the same size and connection probability. A network with a high SWI index is therefore a network with both a high clustering coefficient, similar to a ring lattice network, and small mean path length, similar to a random network.

For directed graphs of size $n$, average degree $k$, and global connection probability $p$:

$C_r = p$ (*Fagiolo, 2007*),

$L_r = \frac{ln(n) - \gamma}{ln(k)} + 0.5$ (*Fronczak et al., 2004*),

$C_l = \frac{3(k-2)}{4(k-1)}$ (*Neal, 2015*)

$L_l = \frac{n}{2k} + 0.5$ (*Neal, 2015*; *Fronczak et al., 2004*)

where $\gamma$ is the Euler-Mascheroni constant.

## Active cluster analysis

To quantify cluster activation (*Figure 5*), we calculated the population rate for each cluster individually as the mean firing rate of all excitatory cells belonging to the cluster smoothed with a Gaussian kernel (15 ms standard deviation). A cluster was defined as 'active' if at any point its population rate exceeded twice that of any other cluster during a PBE. The active clusters' duration of activation was defined as the duration for which it was the most active cluster.

To test whether the sequence of activation in events with three active clusters matched the sequence of place fields on the track, we performed a bootstrap significance test (*Figure 5—figure supplement 1*). For all events from the fiducial parameter set that had three active clusters, we calculated the fraction in which the sequence of the active clusters matched the sequence of the clusters' left vs right bias on the track in either direction. We then compared this fraction to the distribution expected from randomly sampling sequences of three clusters without replacement.

To determine if there was a relationship between the number of active clusters within an event and it's preplay quality, we performed a Spearman's rank correlation between the number of active clusters and the normalized absolute weighted correlation across all events at the fiducial parameter set. The absolute weighted correlations were z-scored based on the absolute weighted correlations of the time-bin shuffled events that had the same number of active clusters.

## Experimental data

Electrophysiological data was reanalyzed from the hippocampal CA1 recordings first published in *Shin et al., 2019*. All place-field data (*Figure 3a*) came from the six rats' first experience on the W-track spatial alternation task. All preplay data (*Figure 4a and b*) came from the six rats' first sleep-box session, which lasted 20–30 min and occurred immediately before their first experience on the W-track.

## Code

Simulations and analysis were performed in MATLAB with custom code. Code available at https://github.com/primon23/Preplay_paper, copy archived at *Miller, 2024*.

## Acknowledgements

NIH/NINDS R01NS104818, NIH/NIMH R01MH112661, NIH/NIMH R01MH120228, and Brandeis University Neuroscience Graduate Program.

---

## Additional information

### Funding

| Funder | Grant reference number | Author |
|---|---|---|
| National Institutes of Health | R01NS104818 | Jordan Breffle<br>Hannah Germaine<br>Paul Miller |
| National Institutes of Health | R01MH112661 | Justin D Shin<br>Shantanu P Jadhav |
| National Institutes of Health | R01MH120228 | Shantanu P Jadhav<br>Justin D Shin |
| Brandeis University | Neuroscience Graduate Program | Jordan Breffle<br>Hannah Germaine |

The funders had no role in study design, data collection and interpretation, or the decision to submit the work for publication.

### Author contributions

Jordan Breffle, Data curation, Software, Formal analysis, Investigation, Visualization, Methodology, Writing - original draft, Writing - review and editing; Hannah Germaine, Software, Investigation, Visualization, Methodology, Writing - review and editing; Justin D Shin, Data curation, Investigation, Methodology, Writing - review and editing; Shantanu P Jadhav, Resources, Supervision, Funding acquisition, Methodology, Project administration, Writing - review and editing; Paul Miller, Conceptualization, Supervision, Funding acquisition, Project administration, Writing - review and editing

### Author ORCIDs

Jordan Breffle ⓘ https://orcid.org/0000-0001-5793-4427
Hannah Germaine ⓘ https://orcid.org/0000-0002-7624-2431
Justin D Shin ⓘ https://orcid.org/0000-0002-7959-7772
Shantanu P Jadhav ⓘ https://orcid.org/0000-0001-5821-0551
Paul Miller ⓘ https://orcid.org/0000-0002-9280-000X

### Ethics

This study was performed in strict accordance with the recommendations in the Guide for the Care and Use of Laboratory Animals of the National Institutes of Health. All of the animals were handled according to approved institutional animal care and use committee (IACUC) protocol #24001-A of Brandeis University. All surgery was performed under ketamine, xylazine, and isoflurane anesthesia, and every effort was made to minimize suffering.

Reviewer #1 (Public review): https://doi.org/10.7554/eLife.93981.3.sa1
Reviewer #2 (Public review): https://doi.org/10.7554/eLife.93981.3.sa2
Reviewer #3 (Public review): https://doi.org/10.7554/eLife.93981.3.sa3
Author response https://doi.org/10.7554/eLife.93981.3.sa4

---

## Additional files

### Supplementary files
• MDAR checklist

### Data availability

All computer codes, which can reproduce all simulated data and carry out analyses can be found at GitHub, copy archived at *Miller, 2024*. The experimental data has been deposited at DANDI Archive.

The following dataset was generated:

| Author(s) | Year | Dataset title | Dataset URL | Database and Identifier |
| --- | --- | --- | --- | --- |
| Shin JD, Jadhav SP | 2024 | Single-day W-track learning | https://doi.org/10.48324/dandi.000978/0.240511.0307 | DANDI Archive, 10.48324/dandi.000978/0.240511.0307 |

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
