## [Editor Report · eLife assessment]

This study presents an **important** finding on the spontaneous emergence of structured activity in artificial neural networks endowed with specific connectivity profiles. The evidence supporting the claims of the authors is **convincing**, providing direct comparison between the properties of the model and neural data although investigating more naturalistic inputs to the network would have strengthened the main claims. The work will be of interest to systems and computational neuroscientists studying the hippocampus and memory processes.

---

## [Referee Report · Reviewer #1 (Public review)]

Summary:

An investigation of the dynamics of a neural network model characterized by sparsely connected clusters of neuronal ensembles. The authors found that such a network could intrinsically generate sequence preplay and place maps, with properties like those observed in the real-world data.

Strengths:

Computational model and data analysis supporting the hippocampal network mechanisms underlying sequence preplay of future experiences and place maps.

The revised version of the manuscript addressed all my comments and as a result is significantly improved.

Weaknesses:

None noted

---

## [Referee Report · Reviewer #2 (Public review)]

Summary:

The authors show that a spiking network model with clustered connectivity produces intrinsic spike sequences when driven with an ramping input, which are recapitulated in the absence of input. This behavior is only seen for some network parameters (neuron cluster participation and number of clusters in the network), which correspond to those that produce a small world network. By changing the strength of ramping input to each network cluster, the network can show different sequences.

Strengths:

A strength of the paper is the direct comparison between the properties of the model and neural data.

Weaknesses:

My main critique of the paper relates to the form of the input to the network. Specifically, it's unclear how much the results depend on the choice of a one-dimensional environment with ramping input. While this is an elegant idealization that allows the authors to explore the representation and replay properties of their model, it is a strong and highly non-physiological constraint. In order to address this concern, the authors would need to test the spatial tuning of their network in 2-dimensional environments, and with different kinds of input from a population of neurons that have a range of degree of spatial tuning and physiological plausibility. A method for systematically producing input with varying degrees of spatial tuning in both 1D and 2D environments has been previously used in (Fang et al 2023, eLife, see Figures 4 and 5), which could be readily adapted for the current study; and behaviorally plausible trajectories in 2D can be produced using the RatInABox package (George et al 2022, bioRxiv), which can also generate e.g. grid cell-like activity that could be used as physiologically plausible input to the network.

---

## [Referee Report · Reviewer #3 (Public review)]

This work offers a novel perspective to the question of how hippocampal networks can adaptively generate different spatial maps and replays of the corresponding place cells, without any such maps pre-existing in the network architecture or its inputs. And how can these place cells preplay their sequences even before the environment is experienced? Previous models required pre-existing spatial representations to be artificially introduced, limiting their adaptability to new environments. Others depended on synaptic plasticity rules which made remapping slower that what is seen in recordings. In contrast, this modeling study proposes that quickly-adaptive intrinsic spiking sequences (preplays) and spatially tuned spiking (place cells) can be generated in a network through randomly clustered recurrent connectivity. By simulating spatial exploration through border-cell-like synaptic inputs, the model generates place cells for different "environments" without the need to reconfigure its synaptic connectivity or introduce plasticity. By simulating sleep-like random synaptic inputs, the model generates sequential activations of cells, mimicking preplays. These "preplays" require small-world connectivity, so that cell clusters are activated in sequence. Using a set of electrophysiological recordings from CA1, the authors confirm that the modeled place cells and replays share many features with recorded ones.

Many features of the model are thoroughly examined, and conclusions are overall convincing (within the simple architecture of the model). Even though the modeled connectivity applies more closely to CA3, it remains unclear whether CA3 recapitulates the proposed small world architecture.

In any case, the proposal that a small-world-structured, clustered network can generate flexible place cells and replays without the need for pre-configured maps is novel and of potential interest to a wide computational and experimental community.

---

## [Author Response]

The following is the authors’ response to the original reviews.

**Public Reviews:**

**Reviewer #1 (Public Review):**
In this manuscript, the authors investigated the dynamics of a neural network model characterized by sparsely connected clusters of neuronal ensembles. They found that such a network could intrinsically generate sequence preplay and place maps, with properties like those observed in the real-world data. Strengths of the study include the computational model and data analysis supporting the hippocampal network mechanisms underlying sequence preplay of future experiences and place maps.Previous models of replay or theta sequences focused on circuit plasticity and usually required a pre-existing place map input from the external environment via upstream structures. However, those models failed to explain how networks support rapid sequential coding of novel environments or simply transferred the question to the upstream structure. On the contrary, the current proposed model required minimal spatial inputs and was aimed at elucidating how a preconfigured structure gave rise to preplay, thereby facilitating the sequential encoding of future novel environments.In this model, the fundamental units for spatial representation were clusters within the network. Sequential representation was achieved through the balance of cluster isolation and their partial overlap. Isolation resulted in a self-reinforced assembly representation, ensuring stable spatial coding. On the other hand, overlap-induced activation transitions across clusters, enabling sequential coding.This study is important when considering that previous models mainly focused on plasticity and experience-related learning, while this model provided us with insights into how network architecture could support rapid sequential coding with large capacity, upon which learning could occur efficiently with modest modification via plasticity.I found this research very inspiring and, below, I provide some comments aimed at improving the manuscript. Some of these comments may extend beyond the scope of the current study, but I believe they raise important questions that should be addressed in this line of research.(1) The expression 'randomly clustered networks' needs to be explained in more detail given that in its current form risks to indicate that the network might be randomly organized (i.e., not organized). In particular, a clustered network with future functionality based on its current clustering is not random but rather pre-configured into those clusters. What the authors likely meant to say, while using the said expression in the title and text, is that clustering is not induced by an experience in the environment, which will only be later mapped using those clusters. While this organization might indeed appear as randomly clustered when referenced to a future novel experience, it might be non-random when referenced to the prior (unaccounted) activity of the network. Related to this, network organization based on similar yet distinct experiences (e.g., on parallel linear tracks as in Liu, Sibille, Dragoi, Neuron 2021) could explain/configure, in part, the hippocampal CA1 network organization that would appear otherwise 'randomly clustered' when referenced to a future novel experience.

As suggested by the reviewer, we have revised the text to clarify that the random clustering is random with respect to any future, novel environment (lines 111-114 and 710-712).

Lines 111-114: “To reconcile these experimental results, we propose a model of intrinsic sequence generation based on randomly clustered recurrent connectivity, wherein place cells are connected within multiple overlapping clusters that are random with respect to any future, novel environment.”

Lines 710-712: “Our results suggest that the preexisting hippocampal dynamics supporting preplay may reflect general properties arising from randomly clustered connectivity, where the randomness is with respect to any future, novel experience.”

The cause of clustering could be prior experiences (e.g. Bourjaily and Miller, 2011) or developmental programming (e.g. Perin et al., 2011; Druckmann et al., 2014; Huszar et al., 2022), and we have modified lines 116 and 714-718 to state this.

Lines 116: Added citation of “Perin et al., 2011”

Lines 714-718: “Synaptic plasticity in the recurrent connections of CA3 may primarily serve to reinforce and stabilize intrinsic dynamics, which could be established through a combination of developmental programming (Perin et al., 2011; Druckmann et al., 2014; Huszar et al., 2022) and past experiences (Bourjaily and Miller, 2011), rather than creating spatial maps de novo.”

We thank the reviewer for suggesting that the results of Liu et al., 2021 strengthen the support for our modeling motivations. We agree, and we now cite their finding that the hippocampal representations of novel environments emerged rapidly but were initially generic and showed greater discriminability from other environments with repeated experience in the environment (lines 130-134).

Lines 130-134: “Further, such preexisting clusters may help explain the correlations that have been found in otherwise seemingly random remapping (Kinsky et al., 2018; Whittington et al., 2020) and support the rapid hippocampal representations of novel environments that are initially generic and become refined with experience (Liu et al., 2021).”

(2) The authors should elaborate more on how the said 'randomly clustered networks' generate beyond chance-level preplay. Specifically, why was there preplay stronger than the time-bin shuffle? There are at least two potential explanations:(1) When the activation of clusters lasts for several decoding time bins, temporal shuffle breaks the continuity of one cluster's activation, thus leading to less sequential decoding results. In that case, the preplay might mainly outperform the shuffle when there are fewer clusters activating in a PBE. For example, activation of two clusters must be sequential (either A to B or B to A), while time bin shuffle could lead to non-sequential activations such as a-b-a-b-a-b where a and b are components of A and B;(2) There is a preferred connection between clusters based on the size of overlap across clusters. For example, if pair A-B and B-C have stronger overlap than A-C, then cluster sequences A-B-C and C-B-A are more likely to occur than others (such as A-C-B) across brain states. In that case, authors should present the distribution of overlap across clusters, and whether the sequences during run and sleep match the magnitude of overlap. During run simulation in the model, as clusters randomly receive a weak location cue bias, the activation sequence might not exactly match the overlap of clusters due to the external drive. In that case, the strength of location cue bias (4% in the current setup) could change the balance between the internal drive and external drive of the representation. How does that parameter influence the preplay incidence or quality?

Explanation 1 is correct: Our cluster-activation analyses (Figure 5) showed that the parameter values that generate preplay correspond to the parameter regions that support sustained cluster activity over multiple decoding time bins, which led us to the conclusion of the reviewer’s first proposed explanation.

We have now added additional analyses supporting the conclusion that cluster-wise activity is the main driver of preplay rather than individual cell-identity (Figures 6 and 7). In Figure 6 we show that cluster-identity alone is sufficient to produce significant preplay by performing decoding after shuffling cell identity within clusters, and in Figure 7 we show that this result holds true when considering the sequence of spiking activity within population bursts rather than the spatial decoding.

Lines 495-515: The pattern of preplay significance across the parameter grid in Figure 4f shows that preplay only occurs with modest cluster overlap, and the results of Figure 5 show that this corresponds to the parameter region that supports transient, isolated cluster-activation. This raises the question of whether cluster-identity is sufficient to explain preplay. To test this, we took the sleep simulation population burst events from the fiducial parameter set and performed decoding after shuffling cell identity in three different ways. We found that when the identity of all cells within a network are randomly permuted the resulting median preplay correlation shift is centered about zero (t-test 95% confidence interval, -0.2018 to 0.0012) and preplay is not significant (distribution of p-values is consistent with a uniform distribution over 0 to 1, chi-square goodness-of-fit test p=0.4436, chi-square statistic=2.68; Figure 6a). However, performing decoding after randomly shuffling cell identity between cells that share membership in a cluster does result in statistically significant preplay for all shuffle replicates, although the magnitude of the median correlation shift is reduced for all shuffle replicates (Figure 6b). The shuffle in Figure 6b does not fully preserve cell’s cluster identity because a cell that is in multiple clusters may be shuffled with a cell in either a single cluster or with a cell in multiple clusters that are not identical. Performing decoding after doing within-cluster shuffling of only cells that are in a single cluster results in preplay statistics that are not statistically different from the unshuffled statistics (t-test relative to median shift of un-shuffled decoding, p=0.1724, 95% confidence interval of -0.0028 to 0.0150 relative to the reference value; Figure 6c). Together these results demonstrate that cluster-identity is sufficient to produce preplay.

Lines 531-551: While cluster-identity is sufficient to produce preplay (Figure 6b), the shuffle of Figure 6c is incomplete in that cells belonging to more than one cluster are not shuffled. Together, these two shuffles leave room for the possibility that individual cell-identity may contribute to the production of preplay. It might be the case that some cells fire earlier than others, both on the track and within events. To test the contribution of individual cells to preplay, we calculated for all cells in all networks of the fiducial parameter point their mean relative spike rank and tested if this is correlated with the location of their mean place field density on the track (Figure 7). We find that there is no relationship between a cell’s mean relative within-event spike rank and its mean place field density on the track (Figure 7a). This is the case when the relative rank is calculated over the entire network (Figure 7, “Within-network”) and when the relative rank is calculated only with respect to cells with the same cluster membership (Figure 7, “Within-cluster”). However, because preplay events can proceed in either track direction, averaging over all events would average out the sequence order of these two opposite directions. We performed the same correlation but after reversing the spike order for events with a negative slope in the decoded trajectory (Figure 7b). To test the significance of this correlation, we performed a bootstrap significance test by comparing the slope of the linear regression to the slope that results when performing the same analysis after shuffling cell identities in the same manner as in Figure 6. We found that the linear regression slope is greater than expected relative to all three shuffling methods for both the within-network mean relative rank correlation (Figure 6c) and the within-cluster mean relative rank correlation (Figure 6d).

Lines 980-1000:

“Cell identity shuffled decoding

We performed Bayesian decoding on the fiducial parameter set after shuffling cell identities in three different manners (Figures 6 and 7). To shuffle cells in a cluster-independent manner (“Across-network shuffle”), we randomly shuffled the identity of cells during the sleep simulations. To shuffle cells within clusters (“Within-cluster shuffle”), we randomly shuffled cell identity only between cells that shared membership in at least one cluster. To shuffle cells within only single clusters (“Within-single-cluster shuffle”), we shuffled cells in the same manner as the within-cluster shuffle but excluded any cells from the shuffle that were in multiple clusters.

To test for a correlation between spike rank during sleep PBEs and the order of place fields on the track (Figure 7), we calculated for each excitatory cell in each network of the fiducial parameter set its mean relative spike rank and correlated that with the location of its mean place field density on the track (Figure 7a). To account for event directionality, we calculated the mean relative rank after inverting the rank within events that had a negatively sloped decoded trajectory (Figure 7b). We calculated mean relative rank for each cell relative to all cells in the network (“Within-network mean relative rank”) and relative to only cells that shared cluster membership with the cell (“Within-cluster mean relative rank”). We then compared the slope of the linear regression between mean relative rank and place field location against the slope that results when applying the same analysis to each of the three methods of cell identify shuffles for both the within-network regression (Figure 7c) and the within-cluster regression (Figure 7d).”

We also now show that the sequence of cluster-activation in events with 3 active clusters does not match the sequence of cluster biases on the track above chance levels and that events with fewer active clusters have the largest increase in median weighted decode correlation (Figure 5—figure supplement 1), showing that the reviewer’s second explanation is not the case.

Lines 466-477: “The results of Figure 5 suggest that cluster-wise activation may be crucial to preplay. One possibility is that the random overlap of clusters in the network spontaneously produces biases in sequences of cluster activation which can be mapped onto any given environment. To test this, we looked at the pattern of cluster activations within events. We found that sequences of three active clusters were not more likely to match the track sequence than chance (Figure 5—figure supplement 1a). This suggests that preplay is not dependent on a particular biased pattern in the sequence of cluster activation. We then we asked if the number of clusters that were active influenced preplay quality. We split the preplay events by the number of clusters that were active during each event and found that the median preplay shift relative to shuffled events with the same number of active clusters decreased with the number of active clusters (Spearman’s rank correlation, p=0.0019, = -0.13; Figure 5—figure supplement 1b).”

Lines 1025-1044:

“Active cluster analysis

To quantify cluster activation (figure 5), we calculated the population rate for each cluster individually as the mean firing rate of all excitatory cells belonging to the cluster smoothed with a Gaussian kernel (15 ms standard deviation). A cluster was defined as ‘active’ if at any point its population rate exceeded twice that of any other cluster during a PBE. The active clusters’ duration of activation was defined as the duration for which it was the most active cluster.

To test whether the sequence of activation in events with three active clusters matched the sequence of place fields on the track, we performed a bootstrap significance test (Figure 5—figure supplement 1). For all events from the fiducial parameter set that had three active clusters, we calculated the fraction in which the sequence of the active clusters matched the sequence of the clusters’ left vs right bias on the track in either direction. We then compared this fraction to the distribution expected from randomly sampling sequences of three clusters without replacement.

To determine if there was a relationship between the number of active clusters within an event and it’s preplay quality we performed a Spearman’s rank correlation between the number of active clusters and the normalized absolute weighted correlation across all events at the fiducial parameter set. The absolute weighted correlations were z-scored based on the absolute weighted correlations of the time-bin shuffled events that had the same number of active clusters.”

We also now add control simulations showing that without the cluster-dependent bias the population burst events no longer significantly decode as preplay (Figure 4—figure supplement 4e).

(3) The manuscript is focused on presenting that a randomly clustered network can generate preplay and place maps with properties similar to experimental observations. An equally interesting question is how preplay supports spatial coding. If preplay is an intrinsic dynamic feature of this network, then it would be good to study whether this network outperforms other networks (randomly connected or ring lattice) in terms of spatial coding (encoding speed, encoding capacity, tuning stability, tuning quality, etc.)

We agree that this is an interesting future direction, but we see it as outside the scope of the current work. There are two interesting avenues of future work: (1) Our current model does not include any plasticity mechanisms, but a future model could study the effects of synaptic plasticity during preplay on long-term network dynamics, and (2) Our current model does not include alternative approaches to constructing the recurrent network, but future studies could systematically compare the spatial coding properties of alternative types of recurrent networks.

(4) The manuscript mentions the small-world connectivity several times, but the concept still appears too abstract and how the small-world index (SWI) contributes to place fields or preplay is not sufficiently discussed.For a more general audience in the field of neuroscience, it would be helpful to include example graphs with high and low SWI. For example, you can show a ring lattice graph and indicate that there are long paths between points at opposite sides of the ring; show randomly connected graphs indicating there are no local clustered structures, and show clustered graphs with several hubs establishing long-range connections to reduce pair-wise distance.How this SWI contributes to preplay is also not clear. Figure 6 showed preplay is correlated with SWI, but maybe the correlation is caused by both of them being correlated with cluster participation. The balance between cluster overlap and cluster isolation is well discussed. In the Discussion, the authors mention "...Such a balance in cluster overlap produces networks with small-world characteristics (Watts and Strogatz, 1998) as quantified by a small-world index..." (Lines 560-561). I believe the statement is not entirely appropriate, a network similar to ring lattice can still have the balance of cluster isolation and cluster overlap, while it will have small SWI due to a long path across some node pairs. Both cluster structure and long-range connection could contribute to SWI. The authors only discuss the necessity of cluster structure, but why is the long-range connection important should also be discussed. I guess long-range connection could make the network more flexible (clusters are closer to each other) and thus increase the potential repertoire.

We agree that the manuscript would benefit from a more concrete explanation of the small-world index. We have added a figure illustrating different types of networks and their corresponding SWI (Figure 1—figure supplement 1) and a corresponding description in the main text (lines 228-234).

Lines 228-234: “A ring lattice network (Figure 1—figure supplement 1a) exhibits high clustering but long path lengths between nodes on opposite sides of the ring. In contrast, a randomly connected network (Figure 1—figure supplement 1c) has short path lengths but lacks local clustered structure. A network with small world structure, such as a Watts-Strogatz network (Watts and Strogatz, 1998) or our randomly clustered model (Figure 1—figure supplement 1b), combines both clustered connectivity and short path lengths. In our clustered networks, for a fixed connection probability the SWI increases with more clusters and lower cluster participation…”

We note that while our most successful clustered networks are indeed those with small-world characteristics, there are other ways of producing small-world networks which may not show good place fields or preplay. We have modified lines 690-692 to clarify that that statement is specific to our model.

Lines 690-692: “In our clustered network structure, such a balance in cluster overlap produces networks with small-world characteristics (Watts and Strogatz, 1998) as quantified by a small-world index (SWI, Figure 1g; Neal, 2015; Neal, 2017).”

(5) What drives PBE during sleep? Seems like the main difference between sleep and run states is the magnitude of excitatory and inhibitory inputs controlled by scaling factors. If there are bursts (PBE) in sleep, do you also observe those during run? Does the network automatically generate PBE in a regime of strong excitation and weak inhibition (neural bifurcation)?

During sleep simulations, the PBEs are spontaneously generated by the recurrent connections in the network. The constant-rate Poisson inputs drive low-rate stochastic spiking in the recurrent network, which then randomly generates population events when there is sufficient internal activity to transiently drive additional spiking within the network.

During run simulations, the spatially-tuned inputs drive greater activity in a subset of the cells at a given point on the track, which in turn suppress the other excitatory cells through the feedback inhibition.

We have added a brief explanation of this in the text in lines 281-284.

Lines 281-284: “During simulated sleep, sparse, stochastic spiking spontaneously generates sufficient excitement within the recurrent network to produce population burst events resembling preplay (Figure 2d-f)”

(6) Is the concept of 'cluster' similar to 'assemblies', as in Peyrache et al, 2010; Farooq et al, 2019? Does a classic assembly analysis during run reveal cluster structures?

Our clusters correspond to functional assemblies in that cells that share a cluster membership have more-similar place fields and are more likely to reactivate together during population burst events. In the figure to the right, we show for an example network at the fiducial parameter set the Pearson correlation between all pairs of place fields split by whether the cells share membership in a cluster (blue) or do not (red).

**Author response image 1. sa4fig1:** 

We expect an assembly analysis would identify assemblies similarly to the experimental data, but we see this additional analysis as a future direction. We have added a description of this correspondence in the text at lines 134-137.

Lines 134-137: “Such clustered connectivity likely underlies the functional assemblies that have been observed in hippocampus, wherein groups of recorded cells have correlated activity that can be identified through independent component analysis (Peyrache et al., 2010; Farooq et al., 2019).”

(7) Can the capacity of the clustered network to express preplay for multiple distinct future experiences be estimated in relation to current network activity, as in Dragoi and Tonegawa, PNAS 2013?

We agree this is an interesting opportunity to compare the results of our model to what has been previously found experimentally. We report here preliminary results supporting this as an interesting future direction.

**Author response image 2. sa4fig2:** 

We performed a similar analysis to that reported in Figure 3C of Dragoi and Tonegawa, 2013. We determined the statistical significance of each event individually for each of the two environments by testing whether the decoded event’s absolute weighted correlation exceeded that 99th percentile of the corresponding shuffle events. We then fit a linear regression to the fraction of events that were significant for each of the two tracks and that were significant to either of the two tracks (left panel of above figure). We then estimated the track capacity as the number of tracks at the point where the linear regression reached 100% of the network capacity. We find that applying this analysis to our fiducial parameter set returns an estimate of ~8.6 tracks (Dragoi and Tonegawa, 2013, found ~15 tracks).

We performed this same analysis for each parameter point in our main parameter grid (right panel of above figure). The parameter region that produces significant preplay (Figure 4f) corresponds to the region that has a track capacity of approximately 8-25 tracks. In the parameter grid region that does not produce preplay, the estimated track capacity approaches the high values that this analysis would produce when applied to events that are significant only at the false-positive rate. This analysis is based on the assumption that each preplay event would significantly correspond to at least one future event. Interesting interpretation issues arise when applying this analysis to parameter regions that do not produce statistically significant preplay, which we leave to future directions to address.

We note two differences between our analysis here and that in Dragoi and Tonegawa, 2013. First, their track capacity analysis was performed on spike sequences rather than decoded spatial sequences, which is the focus of our manuscript. Second, they recorded rats exploring three novel tracks, while in our manuscript we only simulated two novel tracks, which reduces the accuracy of our linear extrapolation of track capacity.

**Reviewer #2 (Public Review):**
Summary:The authors show that a spiking network model with clustered neurons produces intrinsic spike sequences when driven with a ramping input, which are recapitulated in the absence of input. This behavior is only seen for some network parameters (neuron cluster participation and number of clusters in the network), which correspond to those that produce a small world network. By changing the strength of ramping input to each network cluster, the network can show different sequences.Strengths:A strength of the paper is the direct comparison between the properties of the model and neural data.Weaknesses:My main critiques of the paper relate to the form of the input to the network.First, because the input is the same across trials (i.e. all traversals are the same duration/velocity), there is no ability to distinguish a representation of space from a representation of time elapsed since the beginning of the trial. The authors should test what happens e.g. with traversals in which the animal travels at different speeds, and in which the animal's speed is not constant across the entire track, and then confirm that the resulting tuning curves are a better representation of position or duration.

We thank the reviewer for pointing out this important limitation. We see extensive testing of the time vs space coding properties of this network as a future direction, but we have performed simulations that demonstrate the robustness of place field coding to variations in traversal speeds and added the results as a supplemental figure (Figure 3—figure supplement 1).

Lines 332-336: “To verify that our simulated place cells were more strongly coding for spatial location than for elapsed time, we performed simulations with additional track traversals at different speeds and compared the resulting place fields and time fields in the same cells. We find that there is significantly greater place information than time information (Figure 3—figure supplement 1).

Lines 835-841: “To compare coding for place vs time, we performed repeated simulations for the same networks at the fiducial parameter point with 1.0x and 2.0x of the original track traversal speed. We then combined all trials for both speed conditions to calculate both place fields and time fields for each cell from the same linear track traversal simulations. The place fields were calculated as described below (average firing rate within each of the fifty 2-cm long spatial bins across the track) and the time fields were similarly calculated but for fifty 40-ms time bins across the initial two seconds of all track traversals.”

Second, it's unclear how much the results depend on the choice of a one-dimensional environment with ramping input. While this is an elegant idealization that allows the authors to explore the representation and replay properties of their model, it is a strong and highly non-physiological constraint. The authors should verify that their results do not depend on this idealization. Specifically, I would suggest the authors also test the spatial coding properties of their network in 2-dimensional environments, and with different kinds of input that have a range of degrees of spatial tuning and physiological plausibility. A method for systematically producing input with varying degrees of spatial tuning in both 1D and 2D environments has been previously used in (Fang et al 2023, eLife, see Figures 4 and 5), which could be readily adapted for the current study; and behaviorally plausible trajectories in 2D can be produced using the RatInABox package (George et al 2022, bioRxiv), which can also generate e.g. grid cell-like activity that could be used as physiologically plausible input to the network.

We agree that testing the robustness of our results to variations in feedforward input is important. We have added new simulation results (Figure 4—figure supplement 4) showing that the existence of preplay in our model is robust to variations in the form of input.

Testing the model in a 2D environment is an interesting future direction, but we see it as outside the scope of the current work. To our knowledge there are no experimental findings of preplay in 2D environments, but this presents an interesting opportunity for future modeling studies.

Lines 413-420: To test the robustness of our results to variations in input types, we simulated alternative forms of spatially modulated feedforward inputs. We found that with no parameter tuning or further modifications to the network, the model generates robust preplay with variations on the spatial inputs, including inputs of three linearly varying cues (Figure 4—figure supplement 4a) and two stepped cues (Figure 4—figure supplement 4b-c). The network is impaired in its ability to produce preplay with binary step location cues (Figure 4—figure supplement 4d), when there is no cluster bias (Figure 4—figure supplement 4e), and at greater values of cluster participation (Figure 4—figure supplement 4f).

Finally, I was left wondering how the cells' spatial tuning relates to their cluster membership, and how the capacity of the network (number of different environments/locations that can be represented) relates to the number of clusters. It seems that if clusters of cells tend to code for nearby locations in the environment (as predicted by the results of Figure 5), then the number of encodable locations would be limited (by the number of clusters). Further, there should be a strong tendency for cells in the same cluster to encode overlapping locations in different environments, which is not seen in experimental data.

Thank you for making this important point and giving us the opportunity to clarify. We do find that subsets of cells with identical cluster membership have correlated place fields, but as we show in Figure 9b (original Figure 7b) the network place map as a whole shows low remapping correlations across environments, which is consistent with experimental data (Hampson et al., 1996; Pavlides, et al., 2019).

Our model includes a relatively small number of cells and clusters compared to CA3, and with a more realistic number of clusters, the level of correlation across network place maps should reduce even further in our model network. The reason for a low level of correlation in the model is because cluster membership is combinatorial, whereby cells that share membership in one cluster can also belong to separate/distinct other clusters, rendering their activity less correlated than might be anticipated.

We have added text at lines 627-630 clarifying these points.

Lines 628-631: “Cells that share membership in a cluster will have some amount of correlation in their remapping due to the cluster-dependent cue bias, which is consistent with experimental results (Hampson et al., 1996; Pavlides et al., 2019), but the combinatorial nature of cluster membership renders the overall place field map correlations low (Figure 9b).”

**Reviewer #3 (Public Review):**
Summary:This work offers a novel perspective on the question of how hippocampal networks can adaptively generate different spatial maps and replays/preplays of the corresponding place cells, without any such maps pre-existing in the network architecture or its inputs. Unlike previous modeling attempts, the authors do not pre-tune their model neurons to any particular place fields. Instead, they build a random, moderately-clustered network of excitatory (and some inhibitory) cells, similar to CA3 architecture. By simulating spatial exploration through border-cell-like synaptic inputs, the model generates place cells for different "environments" without the need to reconfigure its synaptic connectivity or introduce plasticity. By simulating sleep-like random synaptic inputs, the model generates sequential activations of cells, mimicking preplays. These "preplays" require small-world connectivity, so that weakly connected cell clusters are activated in sequence. Using a set of electrophysiological recordings from CA1, the authors confirm that the modeled place cells and replays share many features with real ones. In summary, the model demonstrates that spontaneous activity within a small-world structured network can generate place cells and replays without the need for pre-configured maps.Strengths:This work addresses an important question in hippocampal dynamics. Namely, how can hippocampal networks quickly generate new place cells when a novel environment is introduced? And how can these place cells preplay their sequences even before the environment is experienced? Previous models required pre-existing spatial representations to be artificially introduced, limiting their adaptability to new environments. Other models depended on synaptic plasticity rules which made remapping slower than what is seen in recordings. This modeling work proposes that quickly-adaptive intrinsic spiking sequences (preplays) and spatially tuned spiking (place cells) can be generated in a network through randomly clustered recurrent connectivity and border-cell inputs, avoiding the need for pre-set spatial maps or plasticity rules. The proposal that small-world architecture is key for place cells and preplays to adapt to new spatial environments is novel and of potential interest to the computational and experimental community.The authors do a good job of thoroughly examining some of the features of their model, with a strong focus on excitatory cell connectivity. Perhaps the most valuable conclusion is that replays require the successive activation of different cell clusters. Small-world architecture is the optimal regime for such a controlled succession of activated clusters.The use of pre-existing electrophysiological data adds particular value to the model. The authors convincingly show that the simulated place cells and preplay events share many important features with those recorded in CA1 (though CA3 ones are similar).Weaknesses:To generate place cell-like activity during a simulated traversal of a linear environment, the authors drive the network with a combination of linearly increasing/decreasing synaptic inputs, mimicking border cell-like inputs. These inputs presumably stem from the entorhinal cortex (though this is not discussed). The authors do not explore how the model would behave when these inputs are replaced by or combined with grid cell inputs which would be more physiologically realistic.

We chose the linearly varying spatial inputs as the minimal model of providing spatial input to the network so that we could focus on the dynamics of the recurrent connections. We agree our results will be strengthened by testing alternative types of border-like input. We show in Figure 4—figure supplement 4that our preplay results are robust to several variations in the location-cue inputs. However, given that a sub-goal of our model was to show that place fields could arise in locations at which no neurons receive a peak in external input, whereas combining input from multiple grid cells produces peaked place-field like input, adding grid cell input (and the many other types of potential hippocampal input) is beyond the scope of the paper.

Even though the authors claim that no spatially-tuned information is needed for the model to generate place cells, there is a small location-cue bias added to the cells, depending on the cluster(s) they belong to. Even though this input is relatively weak, it could potentially be driving the sequential activation of clusters and therefore the preplays and place cells. In that case, the claim for non-spatially tuned inputs seems weak. This detail is hidden in the Methods section and not discussed further. How does the model behave without this added bias input?

We apologize for a lack of clarity if we have caused confusion about the type of inputs and if we implied an absence of spatially-tuned information in the network. In order for place fields to appear the network must receive spatial information, which we model as linearly-varying cues and illustrate in Figure 1b and describe in the caption (original lines 156-157), Results (original lines 189-190 & 497-499), and Methods (original lines 671-683). Such input is not place-field like, as the small bias to any cell linearly decreases from one boundary of the track or the other.

The cluster-dependent bias, which is also described in the same lines (Figure 1 caption (original lines 156-157), Results (original lines 189-190 & 497-499), and Methods (original lines 671-683)), only affects the strength of the spatial cues that are present during simulated run periods. Crucially, this cluster-dependent bias is absent during sleep simulations when preplay occurs, which is why preplay can equally correlate with place field sequences in any context.

We have modified the text (lines 207-210, 218, and 824-827) to clarify these points. We have also added results from a control simulation (Figure 4—figure supplement 4e) showing that preplay is not generated in the absence of the cluster-dependent bias.

Lines 207-210: “This bias causes cells that share cluster memberships to have more similar place fields during the simulated run period, but, crucially, this bias is not present during sleep simulations so that there is no environment-specific information present when the network generates preplay.”

Lines 218: “Second, to incorporate cluster-dependent correlations in place fields, a small…”

Lines 824-827: “The addition of this bias produced correlations in cells’ spatial tunings based on cluster membership, but, importantly, this bias was not present during the sleep simulations, and it did not lead to high correlations of place-field maps between environments (Figure 9b).”

Unlike excitation, inhibition is modeled in a very uniform way (uniform connection probability with all E cells, no I-I connections, no border-cell inputs). This goes against a long literature on the precise coordination of multiple inhibitory subnetworks, with different interneuron subtypes playing different roles (e.g. output-suppressing perisomatic inhibition vs input-gating dendritic inhibition). Even though no model is meant to capture every detail of a real neuronal circuit, expanding on the role of inhibition in this clustered architecture would greatly strengthen this work.

This is an interesting future direction, but we see it as outside the scope of our current work. While inhibitory microcircuits are certainly important physiologically, we focus here on a minimal model that produces the desired place cell activity and preplay, as measured in excitatory cells. We have added a brief discussion of this to the manuscript.

Lines 733-739: “Additionally, the *in vivo* microcircuitry of CA3 is complex and includes aspects such as nonlinear dendritic computations and a variety of inhibitory cell types (Rebola et al., 2017). This microcircuitry is crucial for explaining certain aspects of hippocampal function, such as ripple and gamma oscillogenesis (Ramirez-Villegas et al., 2017), but here we have focused on a minimal model that is sufficient to produce place cell spiking activity that is consistent with experimentally measured place field and preplay statistics.”

For the modeling insights to be physiologically plausible, it is important to show that CA3 connectivity (which the model mimics) shares the proposed small-world architecture. The authors discuss the existence of this architecture in various brain regions but not in CA3, which is traditionally thought of and modeled as a random or fully connected recurrent excitatory network. A thorough discussion of CA3 connectivity would strengthen this work.

We agree this is an important point that is missing, and we have modified lines 114-116 to address the clustered connectivity reported in CA3.

Lines 114-116: “Such clustering is a common motif across the brain, including the CA3 region of the hippocampus (Guzman et al., 2016) as well as cortex (Song et al., 2005), …”

**Recommendations for the authors:**

**Reviewer #1 (Recommendations For The Authors):**
(1) Based on Figure 3, the place fields are not uniformly distributed in the maze. Meanwhile, based on Figure 1b and Methods, the total input seems to be uniform across the maze. Why does the uniform total external input lead to nonuniform network activities?

While the total input to the network is constant across the maze, the input to any individual cell can peak only at either end of the track. All excitatory cells receive input from both the left-cue and the right-cue with different input strengths. By chance and due to the cluster-dependent bias some cells will have stronger input from one cue than the other and will therefore be more likely to have a place field toward that side of the track. However, no cell receives a peak of input in the center of the track. We have modified lines 141-143 to clarify this.

Lines 141-143: “While the total input to the network is constant as a function of position, each cell only receives a peak in its spatially linearly varying feedforward input at one end of the track.”

(2) I find these sentences confusing: "...we expected that the set of spiking events that significantly decode to linear trajectories in one environment (Figure 4) should decode with a similar fidelity in another environment..." (Lines 513-515) and "As expected... but not with the place fields of trajectories from different environments (Figure 7c)" (Line 517-520). What is the expectation for cross-environment decoding? Should they be similar or different? Also, in Figure 7c, the example is not fully convincing. In the figure caption, it states that decoding is significant in the top row but not in the bottom row, but they look similar across rows.

Original lines 513-515 refer to the entire set of events, while original lines 517-520 refer to one example event. The sleep events are simulated without any track-specific information present, so the degree to which preplay occurs when decoding based on the place fields of a specific future track should be independent of any particular track when considering the entire set of decoded PBEs, as shown in Figure 9d (original Figure 7). However, because there is strong remapping across tracks (Figure 9b), an individual event that shows a strong decoded trajectory based on the place fields of one track (Figure 9c, top row) should show chance levels of a decoded trajectory when decoded with the place fields of an alternative track (Figure 9c, bottom row).

We have revised lines 643-650 for clarity, and we have added statistics for the events shown in Figure 9c.

Lines 644-651: “Since the place field map correlations are high for trajectories on the same track and near zero for trajectories on different tracks, any individual event would be expected to have similar decoded trajectories when decoding based on the place fields from different trajectories in the same environment and dissimilar decoded trajectories when decoding based on place fields from different environments. A given event with a strong decoded trajectory based on the place fields of one environment would then be expected to have a weaker decoded trajectory when decoded with place fields from an alternative environment (Figure 9c).

Lines 604-608: “(c) An example event with a statistically significant trajectory when decoded with place fields from Env. 1 left (absolute correlation at the 99th percentile of time-bin shuffles) but not when decoded with place fields of the other trajectories (78th, 45th, and 63rd percentiles, for Env. 1 right, Env. 2 left, and Env. 2 right, respectively). shows a significant trajectory when it is decoded with place fields from one environment (top row), but not when it is decoded with place fields from another environment (bottom row). “

(3) In Methods, the equation at line 610, E in the last term should be E_ext.

We modeled the feedforward inputs as excitatory connections with the same reversal potential as the recurrent excitatory connections, so is the proper value.

(4) Equation line 617 states that conductances follow exponential decay, but the initial conductances of g_I.g_E and g_SRA are not specified.

We have added a description of the initial values in lines 760-764.

Lines 760-764: “Initial feed-forward input conductances were set to values approximating their steady-state values by randomly selecting values from a Gaussian with a mean of and a standard deviation of Win2rGτE. Initial values of the recurrent conductances and the SRA conductance were set to zero.”

(5) In the parameter table below line 647, W_E-E, W_E-I, and W_I-E are not described in the text.

We have clarified in lines 757-760 that the step increase in conductance corresponds to these parameter values.

Lines 757-760: “A step increase in conductance occurs at the time of each spike by an amount corresponding to the connection strength for each synapse (for E-to-E connections, for E-to-I connections, and for I-to-E connections), or by for”.

(6) On line 660, "...Each environment and the sleep session had unique context cue input weights...". Does that mean that within a sleep session, the network received the same context input? How strongly are the sleep dynamics driven by that context input rather than by intrinsic dynamics? Usually, sleep activity is high dimensional, what would happen if the input during sleep is more stochastic?

Yes, within a sleep session each network receives a single set of context inputs, which are implemented as independent Poisson spike trains (so being independent, in small time-windows the dimensionality is equal to the number of neurons). The effects of any particular set of sleep context cue inputs should be minor, since the standard deviation of the input weights, , is small. Further, because the preplay analysis is performed across many networks at each parameter point, the observation of preplay is independent of any particular realization of either the recurrent network or the sleep context inputs.

Further exploring the effects of more biophysically realistic neural dynamics during simulated sleep is an interesting future direction.

(7) One bracket is missing in the denominator in line 831.

We have fixed this error.

Line 1005: “(𝐶_𝑙_ − 𝐶_𝑟_)” -> “(𝐶_𝑙_ − 𝐶_𝑟_)”

**Reviewer #2 (Recommendations For The Authors):**
- I would suggest the authors cite Chenkov et al 2017, PLOS Comp Bio, in which "replay" sequences were produced in clustered networks, and discuss how their work differs.

We have included a contrast of our model to that of Chenkov et al., 2017 in lines 73-78.

Lines 73-78: “Related to replay models based on place-field distance-dependent connectivity is the broader class of synfire-chain-like models. In these models, neurons (or clusters of neurons) are connected in a 1-dimensional feed-forward manner (Diesmann et al., 1999; Chenkov et al., 2017). The classic idea of a synfire-chain has been extended to included recurrent connections, such as by Chenkov et al., 2017, however such models still rely on an underlying 1-dimensional sequence of activity propagation.”

- Figure legend 2e says "replay", should be "preplay".

We have fixed this error.

Line 255: “(e) Example preplay event…”

- How much does the context cue affect the result? e.g. Is sleep notably different with different sleep context cues?

As discussed above in our response to Reviewer 1, the context cue weights have a small standard deviation, , which means that differences in the effects of different realizations of the context inputs are small. Different sets of context cues will cause cells to have slightly higher or lower spiking rates during sleep simulations, but because there is no correlation between the sleep context cue and the place field simulations there should be no effect on preplay quality.

- Figure 4 should include a control with a single cluster.

We thank the reviewer for this suggestion and have added additional control simulations.

In our model, the recurrent structure of a network with a single cluster is equivalent to a cluster-less random network. Additionally, any network where cluster participation equals the number of clusters is equivalent to a cluster-less random network, since all neurons belong to all clusters and can therefore potentially connect to any other neuron. Such a condition corresponds to a diagonal boundary where the number of clusters equals the cluster participation, which occurs at higher values of cluster participation than we had shown in our primary parameter grid.

We now include simulation results that extend to this boundary, corresponding to cluster-less networks (Figure 4—figure supplement 4f). Networks at these parameter points do not show preplay. See our earlier response for the new text associated with Figure 4—figure supplement 4.

- The results of Figure 4 are very noisy. I would recommend increasing the sampling, both in terms of the number of population events in each condition and the number of conditions.

We have run simulations for longer durations (300 seconds) and with more networks (20) to produce more accurate empirical values for the statistics calculated across the parameter grids in Figures 3 and 4. Our additional simulations (Figure 4—figure supplement 4) provide support that the parameter region of preplay significance is reliable.

Lines 831-833: “For the parameter grids in Figures 3 and 4 we simulated 20 networks with 300 s long sleep sessions in order to get more precise empirical estimates of the simulation statistics.”

- It's not entirely clear what's different between the analysis described in lines 334-353, and the preplay analysis in Figure 2. In general, the description of this result was difficult to follow, as it included a lot of text that would be better served in the methods.

In Figure 2 we first introduce the Bayesian decoding method, but it is not until Figure 4 that the shuffle-based significance testing is first introduced. We have simplified the description of the shuffle comparison in lines 371-375 and now refer the reader to the methods for details.

Lines 371-375: “We find significant preplay in both our reference experimental data set (Shin et al., 2019; Figure 4a, b; see Figure 4—figure supplement 1 for example events) and our model (Figure 4c, d) when analyzed by the same methods as Farooq et al., 2019, wherein the significance of preplay is determined relative to time-bin shuffled events (see Methods). For each detected event we calculated its absolute weighted correlation. We then generated 100 time-bin shuffles of each event, and for each shuffle recalculated the absolute weighted correlation to generate a null distribution of absolute weighted correlations.”

- Many of the figures have low text resolution (e.g. Figure 6).

We have now fixed this.

- How does the clustered small world network compare to e.g. a small world ring network as used in Watts and Strogatz 1998?

As described in our above response to Reviewer 1's fourth point, we have added a supplementary figure (Figure 1—figure supplement 1, with corresponding text) comparing our model with the Watts-Strogatz model.

**Reviewer #3 (Recommendations For The Authors):**

Figure 5 would benefit from a plot of the overlap of activated clusters per event.

In our cluster activation analysis in Figure 5, we defined a cluster as “active” if at any point in the event its population rate was twice that of any other clusters’. We used this definition—which permits no overlap of activated clusters—rather than a definition based on a z-scoring of the rate, because we determined that preplay required periods of spiking dominated by individual clusters.

**Author response image 3. sa4fig3:** 

The choice of such a definition is supported by our observation that most spiking activity within an event is dominated by whichever cluster is most active at each point in time. In the left panel of the above figure we show the distribution of the average fraction of spikes within each event that came from the most active cluster at each point in time. The right panel shows the distribution of the average across time within each event of the ratio of the population activity rate of the most active cluster to the second most active cluster. The data for both panels comes from all events at the fiducial parameter set.

**Author response image 4. sa4fig4:** 

Rather than overlapping at a given moment in time, clusters might have overlap in their probability of being active at some point within an event. We do find that there is a small but significant correlation in cluster co-activation. For each network we calculated the activation correlation across events for each pair of clusters (example network show in the left panel). We compared the distribution of resulting absolute correlations against the values that results after shuffling the correlations between cluster activations (right panel, all correlations for all networks from the fiducial parameter point).

Figures 4e/f are referred to as 4c/d in the text (pg 14).

We have fixed this error.

Lines 400-412: “4c” -> “4e” and “4d” -> “4f”